# One-step efficient generation of dual-function conditional knockout and geno-tagging alleles in zebrafish

**Wenyuan Li[1†], Yage Zhang[1†], Bingzhou Han[1†], Lianyan Li[1], Muhang Li[1], Xiaochan Lu[2], Cheng Chen[2], Mengjia Lu[2], Yujie Zhang[1], Xuefeng Jia[3], Zuoyan Zhu[1], Xiangjun Tong[1], Bo Zhang[1]***

[1]Key Laboratory of Cell Proliferation and Differentiation of the Ministry of Education, Peking University Genome Editing Research Center, College of Life Sciences, Peking University, Beijing, China; [2]Laboratory of Chemical Genomics, School of Chemical Biology and Biotechnology, Peking University Shenzhen Graduate School, Shenzhen, China; [3]Gcrispr (Tianjin) Genetic Technology, Tianjin, China

**Abstract** CRISPR/Cas systems are widely used to knock out genes by inducing indel mutations, which are prone to genetic compensation. Complex genome modifications such as knockin (KI) might bypass compensation, though difficult to practice due to low efficiency. Moreover, no 'two-in-one' KI strategy combining conditional knockout (CKO) with fluorescent gene-labeling or further allele-labeling has been reported. Here, we developed a dual-cassette-donor strategy and achieved one-step and efficient generation of dual-function KI alleles at *tbx5a* and *kctd10* loci in zebrafish *via* targeted insertion. These alleles display fluorescent gene-tagging and CKO effects before and after Cre induction, respectively. By introducing a second fluorescent reporter, geno-tagging effects were achieved at *tbx5a* and *sox10* loci, exhibiting CKO coupled with fluorescent reporter switch upon Cre induction, enabling tracing of three distinct genotypes. We found that LiCl purification of gRNA is critical for highly efficient KI, and preselection of founders allows the efficient germline recovery of KI events.
DOI: https://doi.org/10.7554/eLife.48081.001

***For correspondence:**
bzhang@pku.edu.cn

[†]These authors contributed equally to this work

**Competing interests:** The authors declare that no competing interests exist.

## Introduction

Targeted genome modifications *via* customizable endonucleases such as TALENs or the CRISPR/Cas system have revolutionized the technology of genome editing, through which gene knockouts can be easily generated through nonhomologous end joining (NHEJ)-induced indel mutations in many species, including zebrafish (*Danio rerio*) (*Huang et al., 2011*; *Miller et al., 2011*; *Sander et al., 2011*; *Tesson et al., 2011*; *Bedell et al., 2012*; *Huang et al., 2012*; *Chang et al., 2013*; *Cong et al., 2013*; *Hwang et al., 2013*; *Jao et al., 2013*; *Mali et al., 2013*; *Wang et al., 2013*; *Boch et al., 2014*; *Doudna and Charpentier, 2014*; *Bolukbasi et al., 2015*; *Anderson et al., 2017*). However, indel mutations have been reported to be able to evoke transcriptional adaptation-derived genetic compensation, triggered by mutations bearing a premature termination codon (PTC), leading to complications and confusion in the study of gene functions (*Rossi et al., 2015*; *El-Brolosy et al., 2019*; *Ma et al., 2019*). Alternatively, advanced genome engineering, including conditional knockout (CKO) and gene tagging approaches, which rely on targeted knockin (KI) to introduce exogenous DNA sequences (*e.g.*, *loxP* or fluorescent reporter genes) into a target site, might bypass the interference of the unwanted genetic compensation by adopting properly designed strategies (*El-Brolosy and Stainier, 2017*). However, these strategies are difficult in practice due to their

low efficiency and the need for extensive screening, especially when dealing with multicellular organisms without developed embryonic stem (ES) cell technology. In addition, a 'two-in-one' dual-function KI strategy for achieving CKO coupled with gene tagging has rarely been reported (*Su et al., 2017*). Furthermore, the target gene is usually labeled by a single fluorescent reporter, and the concept and method for discriminating different alleles (normal *vs* defective) of the same target gene have not been established.

In zebrafish, KI is generally mediated by either homologous recombination (HR) or the NHEJ pathway for repairing DNA double-strand breaks (*Zu et al., 2013*; *Auer et al., 2014*; *Irion et al., 2014*; *Shin et al., 2014*; *Li et al., 2015a*; *Li et al., 2015b*; *Carroll, 2016*; *Hoshijima et al., 2016*; *Luo et al., 2018*), and NHEJ-based targeted insertion is more efficient, though less precise, than the HR-based strategy (*Hagmann et al., 1998*; *Mao et al., 2008*; *Dai et al., 2010*). However, current NHEJ-based KI methods are feasible for gene labeling but cannot be used to generate CKO alleles through a one-step single insertion since two *loxP* sequences have to be inserted at two separate locations (*Burg et al., 2018*). Alternatively, conditional gene inactivation or in combination with gene tagging can be achieved via various designs of gene-trap cassettes in zebrafish (*Sivasubbu et al., 2006*; *Trinh et al., 2011*; *Ni et al., 2012*; *Sugimoto et al., 2017*; *Grajevskaja et al., 2018*). Nevertheless, CKO and gene labeling are usually achieved through separate experiments, since they typically use different designs of donor DNA, and dual-function CKO alleles coupled with fluorescent gene labeling generated through targeted insertion have not been described. The combination of CKO and gene/cell labeling within the same allele or even CKO together with genotype/allele labeling, allowing researchers to easily distinguish defective alleles from normal ones by using different fluorescent reporters, could be particularly useful for mosaic analysis and cell fate tracing, especially for zebrafish research due to the superiority of the live imaging of embryogenesis in this animal model.

Here, we report the development of a highly efficient method for achieving one-step heritable generation of dual-function alleles in zebrafish through targeted insertion induced by the Cas9/gRNA system, based on the special design of a dual-cassette-donor. The feasibility of this strategy was first demonstrated at the *tbx5a* and *kctd10* loci, resulting in a germline transmission efficiency as high as 56% after the preselection of $F_0$ embryos showing correct expression of a fluorescent reporter. Under normal conditions, these KI alleles enable simultaneous expression of a fluorescent reporter together with the target gene, displaying a gene-labeling effect. After the injection of *Cre* mRNA, the expression and function of the target gene as well as the reporter gene were disrupted, and conditional knockout was successfully achieved. We further improved our strategy by introducing a second fluorescent reporter into the dual-cassette donor, which could be used to conditionally generate and distinguish different alleles (normal *vs* defective) by using different fluorescent reporters and, further, to distinguish different cells with different genotypes (homozygous normal, heterozygote, and homozygous mutant/defective), allowing real-time mosaic analysis within the same tissue. This allele-tagging or geno-tagging strategy was demonstrated at the *tbx5a* and *sox10* loci.

## Results

### Rationale of our dual-cassette-donor strategy for the one-step generation of 'two-in-one' dual-function KI alleles through targeted insertion

To generate a dual-function allele through a single step of NHEJ-mediated KI, we developed a novel strategy based on the special design of a dual-cassette donor, in which CKO and gene labeling can be established simultaneously through one-step targeted insertion of the donor vector into an intron of the target gene *via* the NHEJ pathway activated by the CRISPR/Cas system. For this purpose, the initial donor was designed to contain two functional cassettes, a positive one and a negative one and is therefore called a PoNe (positive-negative) donor (*Figure 1A*). The positive cassette (Po-cassette) was designed to be able to maintain the correct expression/splicing and function of the target gene after the integration of the PoNe donor into the target site, and additionally to track target gene expression through the incorporation of a 2A-linked fluorescent reporter gene. For this purpose, the first part of the Po-cassette was designed to retain the integrity of the cDNA of the target gene by cloning the sequence of the intron, including the splice acceptor site, downstream of the

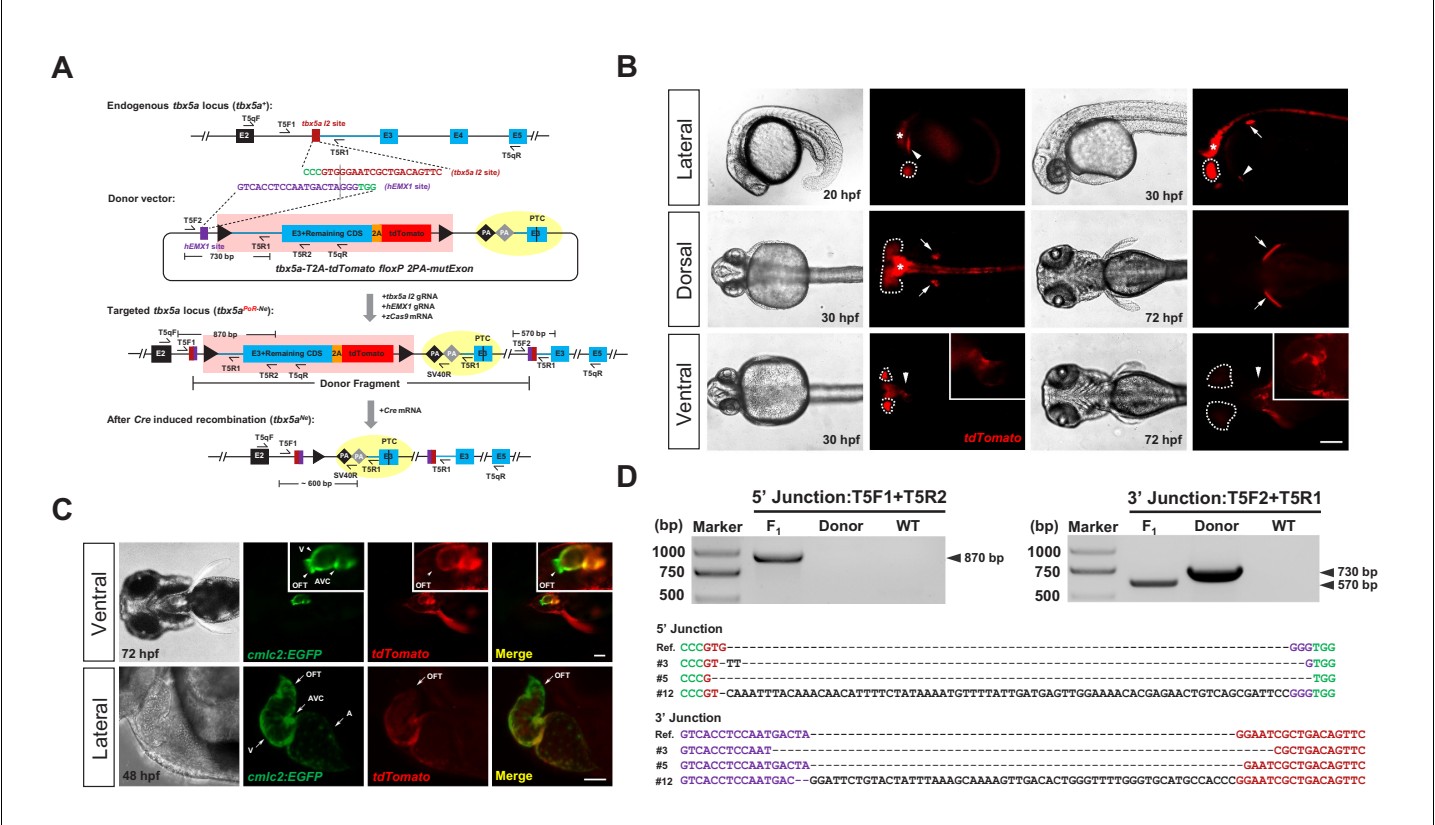

**Figure 1.** Generation of CKO coupled with gene labeling dual-function alleles through targeted insertion in zebrafish at the *tbx5a* locus. (A) Schematic diagram of the KI strategy based on the dual-cassette PoNe donor *tbx5a-T2A-tdTomato floxP 2PA-mutExon* (the *tbx5a* PoR-Ne donor), consisting of a Po-cassette and a Ne-cassette (highlighted by pink and yellow shadows, respectively). The target sequences of *hEMX1* and *tbx5a* are shown in purple and brown, respectively, and the PAMs are in shown green. Black triangles represent *loxP*. Black and gray diamonds indicate polyadenylation (*PA*) signals. The black bar in the third exon (E3) indicates the in-frame premature termination codon (PTC). Primers *T5qF* and *T5qR* are used for qRT-PCR in *Figure 2—figure supplement 1E and F*. (B) Images of $F_1$ larvae from an outcross of a *tbx5a* PoR-Ne donor knockin founder (#12), showing *tdTomato* expression in the pectoral fins (white arrows), heart (white arrowheads), eyes (white dotted circle) and nervous system (white asterisk). Scale bar, 200 μm. (C) Images of $F_1$ progeny from *Tg(cmlc2:EGFP)* transgenic zebrafish crossed with the *tbx5a* KI founder (#12), showing an antero-posterior gradient of *tbx5a* expression in the ventricle. Upper panel: Ventral view of a 72 hpf embryo. Lower panel: Z-stack confocal images of the heart region from a 48 hpf embryo. A: atrium. AVC: atrioventricular canal. OFT: outflow track. V: ventricle. Scale bar, 50 μm. (D) Junction PCR and direct sequencing results of individual positive $F_1$ progeny from outcrosses of each of the three positive founders (#3, #5 and #12). Due to an extra copy of the *T5R1* primer sequence in the PoR-Ne donor, PCR with the primer pair *T5F2* and *T5R1* targeting the donor plasmid results in a larger fragment than that of the $F_1$ progeny. $F_1$: an $F_1$ embryo from $F_0$ #12. Donor: *tbx5a* PoR-Ne donor plasmid. WT: pooled genomic DNA of five wild-type embryos.
DOI: https://doi.org/10.7554/eLife.48081.002

The following figure supplement is available for figure 1:

**Figure supplement 1.** Evaluation of the expression of *tbx5a* and targeted insertion of the CKO + gene labeling PoNe donor at the *tbx5a l2* target site in founder embryos.
DOI: https://doi.org/10.7554/eLife.48081.003

target site as well as all the rest of the downstream coding sequence of the wild-type target gene into the donor. For gene labeling and the visualization of target gene expression, the second part of the Po-cassette was composed of a fluorescent reporter fused in-frame upstream of the stop codon of the target gene cDNA in the donor, separated by a 2A peptide coding sequence. The whole sequence described above was flanked by two *loxP* sites, so that it can be removed to generate conditional mutations in response to Cre recombinase. The negative cassette (Ne-cassette) was designed to ensure complete disruption of target gene expression and function after removal of the Po-cassette by Cre-induced recombination. This is achieved by two levels of interruption of gene expression, leading to the early termination of either transcription or translation. For this purpose, the initial Ne-cassette consisted of two transcription termination signals (TTSs), the SV40 polyA (*PA*)

sequence followed by BGH *PA*, which were followed by the sequence of the same splice acceptor site as in the Po-cassette and the exon immediately after the target intron, but engineered to contain a nonsense mutation (*mutExon*).

After the insertion of the PoNe donor into the intron of the target gene, the targeted locus will acquire the properties of both a conditional allele and a gene-tagging allele. Under normal conditions, that is in the absence of Cre recombinase, this locus serves a gene-tagging function, and the expression and function of the target gene remain normal; a wild-type protein and a fluorescent reporter protein will be produced concurrently, with the fluorescent reporter recapitulating the expression pattern of the endogenous target gene. After Cre/*loxP*-mediated removal of the Po-cassette, the two polyA signals (*2PA*) in the Ne-cassette could prematurely stop the transcription of the target gene, resulting in a short transcript encoding a truncated and nonfunctional protein, thus achieving CKO. If there is leaky transcription after the polyA signals, the *mutExon* sequence in the Ne-cassette will be transcribed and spliced into the mature mRNA, again resulting in the disruption of the function of the target gene, due to the existence of the premature stop codon in this exon.

## Efficient generation of a dual-function KI allele at the zebrafish *tbx5a* locus

The feasibility of this dual-cassette-donor strategy was first evaluated at the *tbx5a* locus (*Basson et al., 1997*; *Garrity et al., 2002*) in zebrafish. We identified a highly efficient guide RNA targeting the second intron of *tbx5a* (referred to the *tbx5a I2* target site, *Figure 1—figure supplement 1A–1C* and *Supplementary file 1*, *2*) and designed a PoNe donor containing an *hEMX1* target site for in vivo linearization (*Figure 1A*; donor name: *tbx5a-T2A-tdTomato floxP 2PA-mutExon*, abbreviated as *tbx5a* PoR-Ne donor) (*Lin et al., 2014*). After the coinjection of this donor plasmid with the *zCas9* mRNA, *tbx5a I2* gRNA and *hEMX1* gRNA into one-cell-stage zebrafish embryos, mosaic expression of *tdTomato* in the heart and pectoral fins, which are the expected tissues of endogenous *tbx5a* expression, was easily observed (*Figure 1—figure supplement 1D* and *Supplementary file 2*). Junction PCR results demonstrated correct integration of the PoNe donor into the *tbx5a I2* target site (*Figure 1—figure supplement 1E*). Surprisingly, we found that the purification methods applied to the gRNAs after in vitro transcription, especially the gRNA targeting the donor plasmid (the *hEMX1* target site), dramatically affected KI efficiency in our experimental settings. In our initial attempts, we employed ethanol precipitation to purify both gRNAs used for injection and found no *tdTomato* expression in any injected embryos. In contrast, when injected with LiCl-purified gRNAs, more than 10% of the embryos showed the expected red fluorescent signal (*Figure 1—figure supplement 1D* and *Table 1*). Furthermore, we could easily observe the expected fluorescent signals in the founder embryos when we injected the Cas9/gRNA system with the donor gRNA (targeting the *hEMX1* site) purified by LiCl precipitation and the endogenous gene (*tbx5a*)-targeting gRNA purified by ethanol precipitation. However, no fluorescent signal could be detected with the other combination, that is the donor gRNA purified by ethanol precipitation and the *tbx5a*-targeting gRNA purified by LiCl precipitation (*Table 1*). These observations indicate that the gRNA targeting the donor plasmids is more sensitive to the purification method, and suggest that the purity or configuration of this gRNA may influence the efficiency of donor linearization in the zebrafish embryos, while the underlying mechanism of this phenomenon needs to be investigated in the future. Nevertheless, in the following experiments, we adopted all LiCl-purified gRNAs for injection.

**Table 1.** KI efficiency using different gRNA purification methods

| *hEMX1* gRNA purification | *tbx5a I2* gRNA purification | Ratio of fluorescence-positive $F_0$ embryos |
| --- | --- | --- |
| LiCl | LiCl | 18/119 (15.1%) |
| LiCl | Ethanol | 8/103 (7.8%) |
| Ethanol | LiCl | 0/94 (0.0%) |
| Ethanol | Ethanol | 0/43 (0.0%) |

DOI: https://doi.org/10.7554/eLife.48081.004

## Germline transmission and evaluation of the *tbx5a* KI allele

After the injection of the *tbx5a* PoR-Ne donor together with the Cas9/gRNA system into approximately 100 one-cell stage embryos, red fluorescent signals could be detected in 18 of the embryos. These founder ($F_0$) embryos were separated into *tdTomato*-positive and *tdTomato*-negative groups and raised to adulthood for germline transmission analysis. Correct expression patterns of red fluorescent signals were detected in the $F_1$ embryos derived from the outcrossing of 5 out of 9 $F_0$ individuals grown from the *tdTomato*-positive embryos, with mosaicism ranging from 2.4% to 43.8% (*Supplementary file 3*), demonstrating successful and highly efficient germline transmission of the *tbx5a* PoR-Ne donor KI events among the embryos preselected for correct fluorescent reporter expression. Junction PCR and sequencing analysis confirmed the germline transmission of the KI alleles (*Figure 1D*). In contrast, no red fluorescent signal was detected in any $F_1$ progeny from the outcrossing of 40 $F_0$ adults raised from *tdTomato*-negative embryos, indicating a positive correlation of reporter gene expression in the founder embryos with successful and efficient germline transmission of the KI events. We also screened several batches of founders prepared from the injection of ethanol-purified gRNAs together with the *tbx5a* PoR-Ne donor and *zCas9* mRNA, and no germline transmission of KI events was detected, demonstrating that LiCl purification of gRNA is critical for the successful generation of heritable KI alleles for our strategy.

Compared with the positive but mosaic founder embryos (*Figure 1—figure supplement 1D*), the $F_1$ progeny from the germline-positive founders showed more specific and uniform expression of *tdTomato* in the heart, pectoral fins, eyes and nervous system (*Figure 1B*). After the introduction of the *Tg(cmlc2:EGFP)* transgenic background (*Tong et al., 2014*), we detected the colocalization of EGFP and tdTomato fluorescent signals in the progeny. Under higher magnification, we found a clear antero-posterior gradient of *tbx5a* expression in the ventricles, with no signal in the outflow track (OFT) and the highest expression in the atrioventricular canal (AVC) (*Figure 1C*). This phenomenon was scarcely detectable through in situ hybridization (*Figure 1—figure supplement 1F*), indicating the advantages of gene labeling for dissecting expression patterns in detail.

## Conditional knockout could be achieved for the *tbx5a* KI allele

The *tbx5a* PoR-Ne donor KI alleles that we generated are not only gene-tagging alleles but also floxed conditional alleles. To evaluate whether conditional knockout could be achieved with these alleles, we isolated a *tbx5a* indel mutation with a 5 bp deletion in the third exon (*tbx5a*$^{\Delta5}$) (*Figure 2—figure supplement 1A–1C*). *tbx5a*$^{\Delta5/\Delta5}$ homozygote embryos showed severe heart and fin defects at 48 hpf (hours post-fertilization) (*Figure 2—figure supplement 1D*), which is consistent with a previous report of a *tbx5a* loss-of-function mutation (*Garrity et al., 2002*). A *tbx5a*$^{+/\Delta5}$ heterozygote was crossed with the *tbx5a* KI founder showing the highest germline mosaicism (#12, 43.8%), and their progeny were injected with *Cre* mRNA at the one-cell stage. As expected, some of the injected $F_1$ embryos (20 out of 78, 25.6%) showed distinct developmental defects of the heart and fins (*Figure 2A*). No red fluorescent signal was observed in any of the injected progeny, indicating that the Cre/*loxP* system was both functional and highly efficient in removing the Po-cassette. In contrast, 30 uninjected embryos developed normally, and 13 of them showed *tdTomato* expression, close to expectations. The genotyping results also showed that all the injected embryos with heart and fin defects were initially *tbx5a*$^{\Delta5/PoR-Ne}$ compound heterozygotes before injection (we designated the original allele bearing the PoR-Ne donor insertion at the *tbx5a* locus from the #12 founder as *tbx5a*$^{PoR-Ne}$, and the converted allele after Cre-induced *loxP* recombination was referred to as *tbx5a*$^{Ne}$), while the normal ones initially presented the *tbx5a*$^{+/\Delta5}$, *tbx5a*$^{+/PoR-Ne}$, or *tbx5a*$^{+/+}$ genotype (*i.e.*, these embryos all harbored at least one wild-type *tbx5a* allele) (*Figure 2B*). To evaluate the expression of the *tbx5a*$^{PoR-Ne}$ KI allele in 'on' and 'off' states at the molecular level, quantitative RT-PCR (qRT-PCR) experiments were performed on 72 hpf embryos using primers *T5qF* and *T5qR* (*Figure 1A*, *Supplementary file 7*) in exons 2 and 5, respectively. The results obtained from outcrosses of *tbx5a*$^{PoR-Ne/PoR-Ne}$ homozygous fish with wild-type fish showed that the mRNA transcription level of the *tbx5a* locus in the *tbx5a*$^{+/PoR-Ne}$ heterozygous embryos without *Cre* mRNA injection was comparable with that in the wild-type control embryos, indicating normal *tbx5a* expression of the *tbx5a*$^{PoR-Ne}$ allele in the 'on' state (*Figure 2—figure supplement 1E*). No red fluorescence signal could be detected in the *tbx5a*$^{+/PoR-Ne}$ embryos after *Cre* mRNA injection, indicating efficient excision of the Po-cassette and that most of the *tbx5a*$^{PoR-Ne}$ alleles had been efficiently converted into

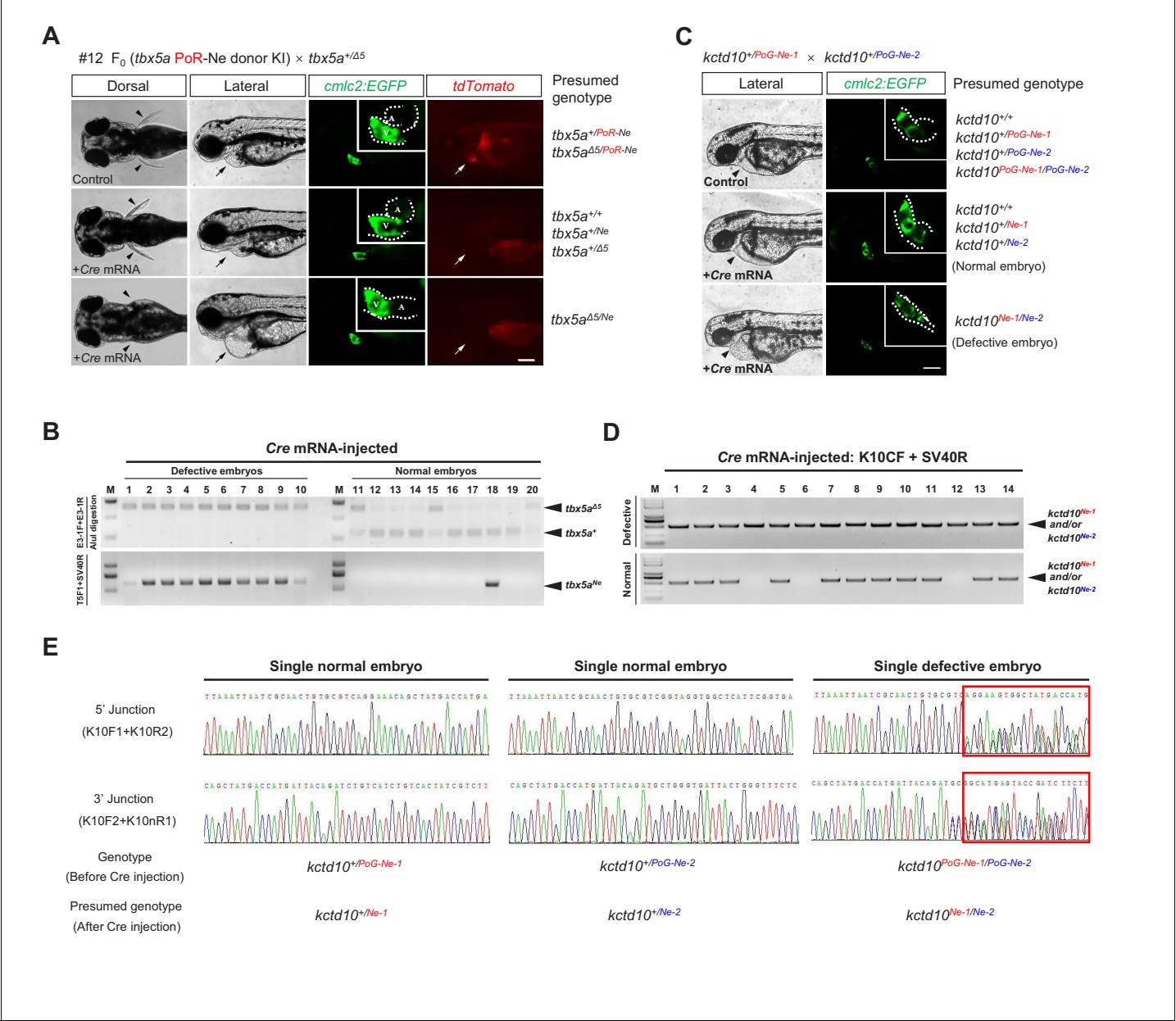

**Figure 2.** The *tbx5a* and *kctd10* conditional alleles are responsive to Cre recombinase. (**A**) Images of the 72 hpf progeny (F$_1$) from a *tbx5a$^{+/\Delta5}$* heterozygote crossed with the F$_0$ #12 (mosaic for the *tbx5a$^{PoR-Ne}$* allele) against the *Tg(cmlc2:EGFP)* transgenic background with or without *Cre* mRNA injection. The fluorescent images were obtained in the lateral view. Black arrowheads indicate pectoral fins, and black or white arrows indicate the heart. A: atrium. V: ventricle. Scale bar, 200 μm. (**B**) Genotyping results of the individual *Cre* mRNA-injected embryos obtained from the cross in A. (**C**) Images of the 48 hpf progeny (F$_2$) derived from the cross of two *kctd10* KI heterozygotes (F$_1$), each carrying a different KI allele (*kctd10$^{PoG-Ne-1}$* from #32 and *kctd10$^{PoG-Ne-2}$* from #5) with the *Tg(cmlc2:EGFP)* transgenic background, to reveal the morphology of the heart. The white dotted line indicates the outline of the heart. The hearts in the upper and middle panels developed normally, showing obvious heart looping. In contrast, the heart in the lower panel shows defective development, exhibiting AVC malformations and heart looping failure. Black arrowheads indicate the heart. Scale bar, 200 μm. (**D**) Genotyping by PCR amplification of the region flanking the *loxP* recombination site of the *Cre* mRNA-injected individual embryos obtained from the cross in C. (**E**) Representative junction PCR and direct sequencing results of the *Cre* mRNA-injected individual embryos showing normal or defective heart development obtained from the cross in C. As expected, the results indicate that the embryo showing the heart phenotype (labeled as 'Single defective embryo' in the figure) was a *kctd10$^{PoG-Ne-1/PoG-Ne-2}$* compound heterozygote (F$_2$) before *Cre* mRNA injection since it showed overlapping peaks (red boxed region) in the sequencing results of the PCR products at both the 5' and 3' junctions (right panel), as the two alleles have different indel sequences at the junction sites. In contrast, the normal embryos (labeled as 'Single normal embryos' in the figure) were either *kctd10$^{+/PoG-Ne-1}$* or *kctd10$^{+/PoG-Ne-2}$* heterozygotes before *Cre* mRNA injection and therefore displayed uniform sequencing results corresponding to either the *kctd10$^{PoG-}$*

*Figure 2 continued on next page*

Figure 2 continued

Ne-1 (or kctd10^Ne-1) or kctd10^PoG-Ne-2 (or kctd10^Ne-2) allele, respectively. The expected corresponding sequences can be found in **Figure 2—figure supplement 2J and K**.

DOI: https://doi.org/10.7554/eLife.48081.005

The following figure supplements are available for figure 2:

**Figure supplement 1.** Evaluation of the indel efficiency of the *tbx5a E3* target site and phenotype analysis of the *tbx5a* indel mutation.

DOI: https://doi.org/10.7554/eLife.48081.006

**Figure supplement 2.** Strategy and evaluation of the targeted insertion of the PoG-Ne donor at the *kctd10* locus.

DOI: https://doi.org/10.7554/eLife.48081.007

*tbx5a*^Ne alleles. The *tbx5a* expression level in these *tbx5a*^+/Ne embryos significantly decreased to approximately 50% of the level in the wild-type embryos, as expected (**Figure 2—figure supplement 1E**). To measure the expression of the *tbx5a*^PoR-Ne allele in the 'off' state in homozygous mutants achieved by Cre-induced excision of the Po-cassette, we also performed qRT-PCR experiments on the embryos obtained from outcross of *tbx5a*^PoR-Ne/PoR-Ne homozygous fish with a *tbx5a*^+/PoR-Ne heterozygote after *Cre* mRNA injection. The results showed that the *tbx5a* transcripts decreased by ~15 fold in the *tbx5a*^Ne/Ne homozygous mutant embryos compared with wild-type ones, indicating highly efficient mutagenesis effect of the *tbx5a*^Ne allele (**Figure 2—figure supplement 1F**). Collectively, these results indicate that our PoNe donor knockin strategy can efficiently achieve a conditional gene knockout effect in addition to gene labeling in zebrafish.

## Generation and evaluation of dual-function alleles showing CKO with a gene labeling effect at the zebrafish *kctd10* locus

We generated a second knockin allele according to the same strategy but replacing *tdTomato* with *tdGFP* as the reporter at another zebrafish locus, *kctd10* (**Figure 2—figure supplement 2A–2D**), which has been reported to play an important role in heart development by suppressing Tbx5a activity, although its expression is somewhat ubiquitous (**Tong et al., 2014**). Similarly, we injected approximately 100 one-cell stage embryos and observed the mosaic expression of *tdGFP* in 22 embryos and we further verified the knockin events by junction PCR in the injected $F_0$ embryos (**Figure 2—figure supplement 2E–2F** and **Supplementary file 2**). This time, we raised all the $F_0$ embryos to adulthood without preselection, including those that were negative for the tdGFP fluorescent signals and found that 2 of the 42 $F_0$ (4.8%) displayed germline transmission of the *kctd10* PoG-Ne donor KI events (**Figure 2—figure supplement 2G–2J** and **Supplementary file 4**). *kctd10* has been reported to show strong maternal expression in zebrafish (**Tong et al., 2014**), and this was confirmed with our *kctd10* PoG-Ne donor KI alleles, since the expression of *tdGFP* could be detected as early as at one-cell stage in all the progeny from outcrosses of female KI heterozygotes but not in any of those from outcrosses of male KI heterozygotes. After the injection of the *Cre* mRNA into the progeny from a cross of two *kctd10* KI heterozygotes ($F_1$) derived from two $F_0$ individuals harboring two slightly different PoG-Ne insertions (designated as *kctd10*^PoG-Ne-1 and *kctd10*^PoG-Ne-2, respectively, differing in the 5' and 3' junction sequences) (**Figure 2—figure supplement 2J and K** and **Supplementary file 4**), 25.9% (74/286) of the embryos exhibited the *kctd10* mutant phenotype, and the rest of the injected embryos and all the uninjected embryos developed normally (**Figure 2C**), indicating that the Po-cassettes were efficiently excised from the *kctd10* PoG-Ne KI alleles. The genotyping and junction PCR results confirmed that all the embryos displaying developmental defects showed *loxP* recombination and overlapping peaks in the sequencing results of the junction PCR products (**Figure 2D–2E**), indicating that they were *kctd10*^PoG-Ne-1/PoG-Ne-2 compound heterozygotes before *Cre* mRNA injection. Similar to the *tbx5a* KI allele, qRT-PCR experiments were also performed to evaluate the mRNA expression level of the edited *kctd10* allele with or without Cre-induced Po-cassette excision, using primers K10qF and K10qR (**Figure 2—figure supplement 2D**, **Supplementary file 7**) in exons 1 and 2, respectively. As expected, the results showed no significant difference in the expression of the *kctd10* locus between the wild-type and *kctd10*^+/PoG-Ne-1 heterozygous embryos, while its expression was significantly reduced by approximately 50% in the *kctd10*^+/PoG-Ne-1 embryos after *Cre* mRNA injection due to the efficient conversion of the *kctd10*^PoG-Ne-1 alleles into the *kctd10*^Ne-1 (**Figure 2—figure supplement 2L**). The *kctd10*

transcription level further decreased to about 0.3% in the $kctd10^{Ne-1/Ne-1}$ homozygous mutant embryos obtained by *Cre* mRNA injection of the progeny from the outcross of $kctd10^{PoG-Ne-1/PoG-Ne-1}$ homozygous fish with a $kctd10^{+/PoG-Ne-1}$ heterozygote, demonstrating highly efficient mutagenesis effect of the $kctd10^{Ne-1}$ allele (*Figure 2—figure supplement 2M*).

In the initial experiments, we used a donor with a single SV40 *PA* termination signal sequence without the modified exon (*mutExon*) in its Ne-cassette. Using such donors, we were able to isolate stable integrations at the *kctd10* locus and observed the correct expression pattern of *tdGFP*. However, after *Cre* mRNA injection, all the injected embryos resulting from the incrosses of such heterozygous F$_1$ fish developed normally, although the green fluorescent signal was no longer detectable in most of the embryos, indicating the successful and efficient removal of the Po-cassette. These results suggest that the single SV40 *PA* signal might not be strong enough to efficiently stop target gene transcription; thus, either multiple copies of the transcription termination sequences are necessary, or the modified exon (*mutExon*) following the *PA* sequence might be indispensable for efficiently disrupting target gene expression in our strategy (data not shown).

## Improving the dual-cassette donor strategy to achieve a geno-tagging effect

The above results demonstrated the unique advantages of our dual-cassette donor strategy for efficiently generating dual-function alleles and achieving CKO coupled with gene labeling in zebrafish *via* one-step targeted insertion. To further improve our strategy and achieve more complex applications of genome editing, we developed a novel technique by designing a conditional dual-fluorescence reporter donor (*Figure 3A*), an advanced version of our original PoNe donor, in which Cre-mediated excision of the Po-cassette could simultaneously trigger conditional knockout and the fluorescent reporter switch, resulting in allele tagging instead of gene tagging, so that different alleles (*i.e.*, normal/positive alleles *vs* defective/negative alleles) can be labeled with different fluorescent proteins. This goal was achieved by adding a different fluorescent reporter in the original Ne-cassette following the sequence containing the intron (including the splice acceptor site) plus a partial exon immediately downstream of the intron target site, and linked by a 2A peptide, as in the Po-cassette. After the removal of the Po-cassette containing the first fluorescent reporter by the Cre/*loxP* system, the second fluorescent reporter in the Ne-cassette will be transcribed together with the preceding intron and exon sequences and spliced into the incomplete mRNA of the target gene, thus achieving both target gene mutation (due to the incomplete coding sequence) and labeling of the mutated allele, simultaneously leading to the destruction of the target gene's function and the allele-tagging switch. Under this strategy, the functionally normal and defective alleles can be easily distinguished by using two different fluorescent reporters. Note that the wild-type exon sequence was used in the Ne-cassette in this case since we would like to maintain normal transcription and translation of the fusion mRNA, which contains the incomplete target gene cDNA followed by a 2A-linked complete sequence encoding the fluorescent reporter (*Figure 3A*). In contrast to the previous strategy, which was based on a single fluorescent reporter, we designated this advanced strategy geno-tagging, since it can label different alleles with different fluorescent reporters, and the genotype of each cell is easily distinguishable by its fluorescent signal.

## Generation and evaluation of geno-tagging alleles at the zebrafish *tbx5a* locus

To test the feasibility of this geno-tagging strategy, we constructed a *tbx5a* geno-tagging donor based on the previous PoR-Ne donor, in which *tdGFP* was inserted into the negative cassette as the second fluorescent reporter followed by two additional polyA signals, and replaced the *mutExon* with the wild-type sequence (*Figure 3A*). This vector was referred to as the *tbx5a* PoR-NeG donor. In the generation of the previous $tbx5a^{PoR-Ne}$ KI allele, we showed that the preselection of F$_0$ embryos according to the fluorescent signal largely ensured the successful identification of the germline transmission of the KI events. However, preselection of F$_0$ embryos according to fluorescent signals is only feasible for certain genes and could be difficult to apply for many other genes, such as those showing late, low or little embryonic expression. To solve this problem, we aimed to establish a more universal method for the enrichment of the germline transmission of KI events. Therefore, during the generation of geno-tagging alleles at the *tbx5a* locus, we evaluated the

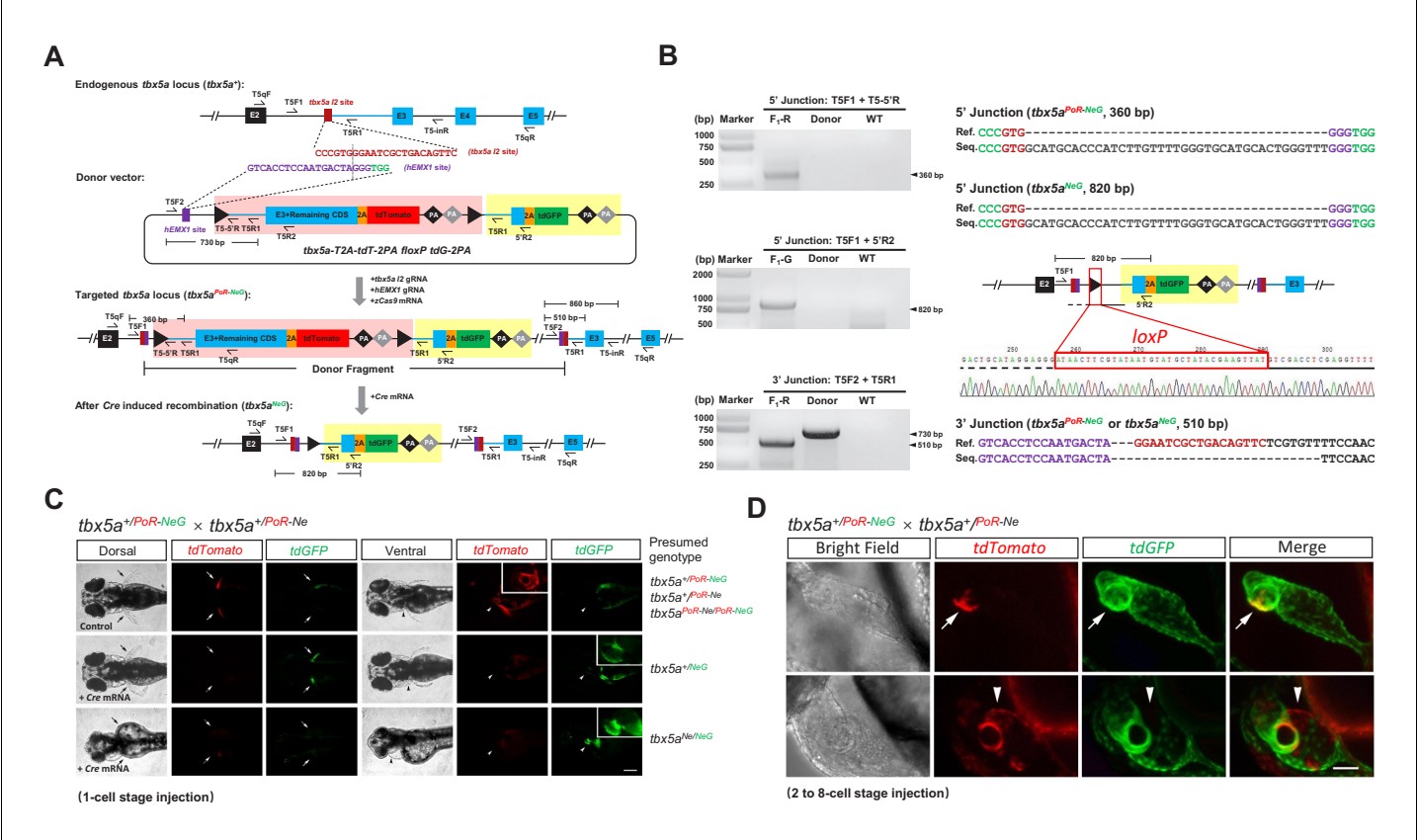

**Figure 3.** Generation of geno-tagging alleles by improving the dual-cassette donor strategy at the zebrafish *tbx5a* locus. (**A**) Schematic diagram of the KI strategy for geno-tagging based on the *tbx5a-T2A-tdT-2PA floxP tdG-2PA* donor (or abbreviated as *tbx5a* PoR-NeG donor). Primers *T5qF* and *T5qR* are used for qRT-PCR in *Figure 3—figure supplement 1E and F*. (**B**) Results of junction PCR and direct sequencing to detect the *tbx5a* geno-tagging donor knockin and Cre-induced recombination events in 48 hpf embryos obtained from the cross in *Figure 3—figure supplement 1B*, that is the $F_1$ embryos from the #42 positive $F_0$ outcrossed with a wild-type zebrafish before or after *Cre* mRNA injection. Note that the sequences of both the *T5F2* and *T5R1* primers are also present in the donor vector, flanking the upstream *loxP* site (as shown in panel A); therefore, a 730 bp product could be amplified in the lane with the donor as the template. $F_1$-R: an $F_1$ embryo showing a red fluorescent signal (before *Cre* mRNA injection). $F_1$-G: an $F_1$ embryo showing a green fluorescent signal (after *Cre* mRNA injection). Donor: *tbx5a* PoR-NeG geno-tagging donor plasmid. WT: pooled genomic DNA of five wild-type embryos. (**C**) Phenotype analysis of the 72 hpf embryos from *tbx5a*$^{+/PoR-NeG}$ heterozygotes (derived from $F_0$ #42 in *Supplementary file 5*) crossed with a *tbx5a*$^{+/PoR-Ne}$ heterozygote (derived from $F_0$ #12 in *Supplementary file 3*) after the injection of *Cre* mRNA at the 1 cell stage. The upper panel represents an uninjected control embryo showing only a red fluorescent signal, whose genotype should be either *tbx5a*$^{+/PoR-Ne}$, *tbx5a*$^{+/PoR-NeG}$ or *tbx5a*$^{PoR-Ne/PoR-NeG}$. The middle panel represents a *Cre* mRNA-injected embryo showing normal development, whose genotype is expected to be *tbx5a*$^{+/NeG}$. The lower panel represents a *Cre* mRNA-injected embryo showing a typical *tbx5a* mutant phenotype, including heart region defects and a lack of pectoral fins, whose genotype is expected to be *tbx5a*$^{Ne/NeG}$. Arrows indicate the pectoral fins, and arrowheads indicate the heart region. The boxed insets show a higher magnification of the corresponding heart region, for better comparison of heart morphology. Scale bar, 200 μm. (**D**) Z-stack confocal images of two representative 48 hpf embryos after the injection of *Cre* mRNA at the 2- to 8 cell stage from the same cross as in C. The white arrows indicate the colocalization of the tdGFP and tdTomato signals, and the white arrowheads indicate the mutually exclusive expression of the tdGFP and tdTomato signals. Scale bar, 50 μm.

DOI: https://doi.org/10.7554/eLife.48081.008

The following figure supplement is available for figure 3:

**Figure supplement 1.** Evaluation of the *tbx5a* geno-tagging effect.
DOI: https://doi.org/10.7554/eLife.48081.009

potential correlation of KI events in germline transmission with the KI efficiency in $F_0$ adult somatic tissues such as caudal fins. After the injection of the *tbx5a* geno-tagging donor (PoR-NeG donor) together with the *zCas9* mRNA and corresponding gRNAs, red fluorescent signals could be detected in 16 of 124 normally developed $F_0$ embryos. The ratio of embryos showing red fluorescent signals was comparable with that in the previous *tbx5a* PoR-Ne donor KI experiment (*Supplementary file 2*). However, we did not select the embryos according to their fluorescent

signals and raised all the $F_0$ embryos together to adulthood in this case. Then, junction PCR analysis was performed using genomic DNA isolated from caudal fin clips from 48 $F_0$ adult fish, and the expected 5′ junction and/or 3′ junction amplicons were detected in six $F_0$ adults (#1, #2, #9, #11, #24 and #42), indicating the successful integration of the donor in somatic cells (*Figure 3—figure supplement 1A*). $F_1$ progeny were obtained from the outcrossing of five such $F_0$ individually with wild-type zebrafish ($F_0$ #24 failed to produce offspring), and germline transmission of the *tbx5a* PoR-NeG donor geno-tagging alleles was detected and confirmed in two $F_0$ individuals (#2 and #42) (*Figure 3B* and *Supplementary file 5*). In contrast, no germline transmission events were detected in the $F_1$ progeny from 20 $F_0$ fish from the junction PCR-negative group. These results indicate a strong correlation between the germline transmission of KI events and adult somatic tissue integration of the donor, and provide an alternative efficient method for the enrichment of germline transmission events.

We chose positive founder #42 for the following experiments and designated the corresponding geno-tagging allele *tbx5a*$^{PoR-NeG}$. We injected *Cre* mRNA into the $F_1$ progeny from $F_0$ #42 outcrossed with wild-type zebrafish and observed successful switching of the fluorescent signals (*Figure 3—figure supplement 1B*). Junction PCR and sequencing results further confirmed the correct KI events and excision of the PoR-cassette. Note that the 3′ junction does not change after *Cre* mRNA injection (*Figure 3B*). To evaluate the geno-tagging effect, we crossed a *tbx5a*$^{+/PoR-NeG}$ heterozygote with a previously identified *tbx5a*$^{+/PoR-Ne}$ heterozygote and divided the embryos into three groups: Group I was injected with 100 pg of *Cre* mRNA at the one-cell stage, Group II was injected with 25 pg of *Cre* mRNA in a single cell at the 2–8 cell stage, and Group III remained untreated as a control (*Figure 3—figure supplement 1C*). No tdTomato signal could be detected in any of the Group I embryos, indicating efficient switching of the fluorescent reporters in all the geno-tagging alleles. Among these embryos, 27.4% (29/106) showed the typical *tbx5a* mutant phenotype (*Figure 3—figure supplement 1C*), and all of these defective embryos were positive for the green fluorescent signal, as expected (*Figure 3C*). Mosaic expression of *tdTomato* and *tdGFP* could be observed in some embryos from Group II, indicating partial switching of the geno-tagging alleles (*Figure 3D*). In some embryos, double fluorescent signals could be observed in the same cell, suggesting that the genotype of these cells was most likely to be *tbx5a*$^{PoR-Ne/NeG}$ (*Figure 3D*). Among the Group II embryos, 11.6% (8/69) also showed obvious developmental defects in the heart and fins (*Figure 3—figure supplement 1C*), and not surprisingly, all of these defective embryos showed dominant or exclusive green fluorescent signals compared with the red fluorescent signal (*Figure 3D*). In contrast, all the embryos from Group III appeared normal during development (*Figure 3—figure supplement 1C*), and only *tdTomato* expression was observed, as expected (*Figure 3C*).

To verify that tissue and temporal control of Cre-induced recombination events can be achieved for this geno-tagging allele, we generated a transgenic line, *Tg(cmlc2:zCreER*$^{T2}$*-2A-ECFP)*, showing heart-specific expression of both *ECFP* and 4-hydroxyltamoxifen (4-HT)-inducible zebrafish codon-optimized *CreER*$^{T2}$. After crossing this transgenic fish with *tbx5a*$^{PoR-NeG/PoR-NeG}$ adults, the embryos were collected and separated into two groups. Then 4-HT was added to one group from 5 hpf to 48 hpf. Green fluorescent signals were detected in the hearts of 56 hpf ECFP-positive embryos, indicating the successful switching of *tdTomato* to *tdGFP* expression in some cells (*Figure 3—figure supplement 1D*). In contrast, only red fluorescent signals were observed in the control group without chemical treatment (data not shown). qRT-PCR experiments were performed to evaluate the expression of the *tbx5a* geno-tagging allele with a similar strategy to that used for the previous *tbx5a*$^{PoR-Ne}$ allele. As expected, the expression of *tbx5a* in the *tbx5a*$^{+/PoR-NeG}$ heterozygous embryos was comparable with the wild-type, but reduced to ~50% after the Cre-induced excision of the Po-cassette (*Figure 3—figure supplement 1E*). Furthermore, mRNA quantification results of the *tbx5a*$^{NeG/NeG}$ homozygous mutant embryos derived from the *Cre* mRNA-injected *tbx5a*$^{PoR-NeG/PoR-NeG}$ embryos showed ~12.5 fold reduction of the *tbx5a* transcripts, confirmed the high mutagenesis efficiency of the *tbx5a*$^{NeG}$ allele (*Figure 3—figure supplement 1F*). These results together demonstrated that our geno-tagging technique successfully achieved switching of fluorescent reporters and CKO simultaneously, which provides unique opportunities for mosaic analysis and lineage tracing of cells with different genotypes.

## Geno-tagging effect achieved for the zebrafish *sox10* gene

To further prove the feasibility of our geno-tagging technique, we also applied this strategy at the *sox10* locus (*Figure 4—figure supplement 1A–1C*), which is mainly expressed in neural crest cells in zebrafish (*Dutton et al., 2001a*; *Dutton et al., 2001b*). It has been reported that the efficiency of transcription termination correlates with the number of polyA signals (*Ni et al., 2012*). We began *sox10* geno-tagging experiments before the work on *tbx5a* and *kctd10* was completed. Because of concern that the double SV40/BGH polyA signals may be insufficient to achieve complete transcriptional termination (*Ni et al., 2012*), we constructed a vector with eight tandem repeats of SV40 pA (*8xPA*) (*Figure 4—figure supplement 1D*). In addition, since the *hEMX1* gRNA targeting the previous donors needs to be purified by LiCl precipitation for efficient KI, we used another highly efficient gRNA target site known as *lamGolden* (*Zu et al., 2016*) for the linearization of the *sox10* geno-tagging donor. However, similar to the case of *hEMX1*, the gRNA targeting the *lamGolden* site also needed to be purified with LiCl for successful knockin; otherwise, knockin events were scarcely detectable in $F_0$ embryos (data not shown). We first coinjected the *sox10* geno-tagging donor with the *zCas9* system into one-cell-stage zebrafish embryos, after which approximately half of the embryos were injected again with 25 pg of *Cre* mRNA in a single cell at the 4 cell stage. As expected, we detected the expression of two different fluorescent reporters in otic vesicles as well as in cranial and trunk neural crest cells in some embryos (29/126) injected with *Cre* mRNA (*Figure 4A*). In some cells, colocalization of the tdGFP and tdTomato signals could be observed (*Figure 4A*), indicating that these cells were most likely to be *sox10*$^{PoR-NeG/NeG}$ heterozygous. In contrast, only the tdTomato signal was observed in the embryos injected without *Cre* mRNA (103/152) (*Supplementary file 2*). NHEJ-mediated knockin and Cre-induced recombination events were further confirmed by junction PCR and sequencing (*Figure 4B*). The 103 founder embryos showing red fluorescent signals were raised to adulthood, and germline screening was conducted outcrossed with wild-type zebrafish. Correct red fluorescent patterns in $F_1$ embryos were observed in 10 out of 21 founders, and the proportion of mosaicism varied from 8.2% to 50.0% (*Supplementary file 6*). Positive $F_1$ embryos from $F_0$ #6 were raised as *sox10*$^{+/PoR-NeG}$ heterozygotes for further study. To test the functionality of the *sox10*$^{PoR-NeG}$ allele, the $F_2$ progeny from an incross of the $F_1$ heterozygotes were injected with 100 pg *Cre* mRNA per embryo at the one-cell stage. An efficient switch of the fluorescent signals from red (tdTomato) to green (tdGFP) was observed in all the injected fluorescent embryos (*Figure 4—figure supplement 1E*), indicating successful high-effciency deletion of the positive cassettes of the *sox10*$^{PoR-NeG}$ alleles in response to Cre recombinase. Furthermore, 24.6% (56/228) of the injected embryos exhibited extensive loss of pigmentation at 48 hpf, recapitulating the phenotype reported previously for the *sox10* mutant *colourless* (*Dutton et al., 2001a*; *Dutton et al., 2001b*), and all these defective embryos exclusively showed green fluorescent signals, indicating that they were most likely homozygous for the *sox10*$^{NeG}$ allele (*Figure 4—figure supplement 1E*). Genotyping results confirmed that all the abnormal embryos were *sox10*$^{PoR-NeG/PoR-NeG}$ homozygotes before *Cre* injection and that all the normal embryos contained at least one wild-type *sox10* allele (*Figure 4—figure supplement 1F*). qRT-PCR experiments were performed to further evaluate the expression of the *sox10* geno-tagging allele. As expected, there was no significant difference in mRNA levels between the *sox10* wild-type allele and the geno-tagging allele without *Cre* mRNA injection, while the expression of the *sox10* geno-tagging allele was significantly decreased after Cre-induced excision of the Po-cassette (*Figure 4—figure supplement 1G*). The transcription level of *sox10* in the *sox10*$^{NeG/NeG}$ homozygous mutant embryos obtained by Cre-induced Po-cassette excision in the *sox10*$^{PoR-NeG/PoR-NeG}$ embryos decreased by ~25 fold compared with wild-type embryos, further proved the strong mutagenesis effect of the *sox10*$^{NeG}$ allele (*Figure 4—figure supplement 1H*). Taken together, these results demonstrated the universality and high efficiency of our geno-tagging strategy.

## Discussion

In summary, we reported a highly efficient knockin strategy based on the special design of multipurpose dual-cassette PoNe donors and NHEJ-mediated targeted insertion, facilitated by the CRISPR/Cas system. Three levels of complex genome modifications could be achieved in zebrafish by using this method: (i) one-step generation of conditional knockout alleles through targeted insertion; (ii) simultaneous fluorescent labeling of CKO alleles to achieve a gene tagging effect as well as labeling

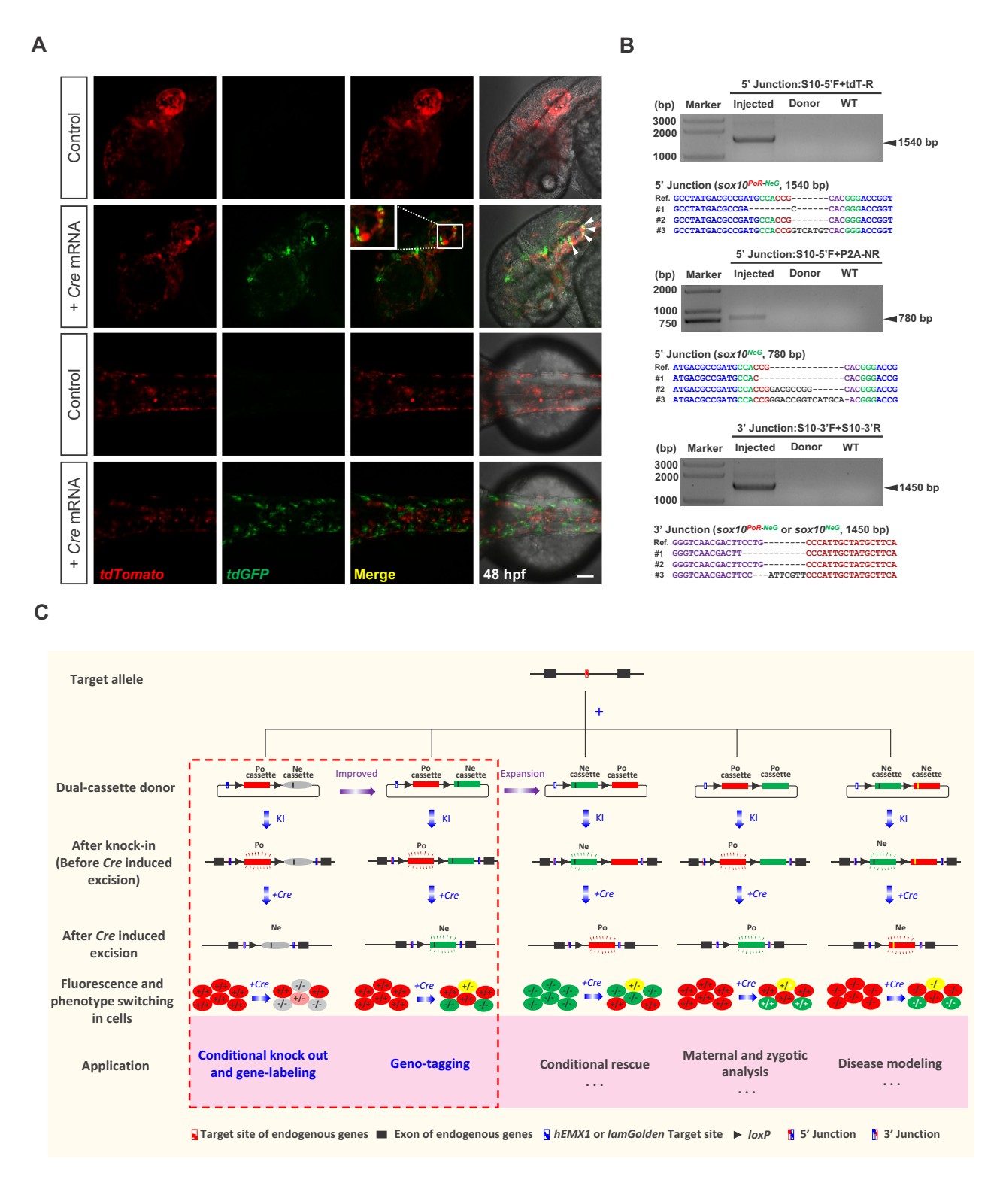

**Figure 4.** Generation of geno-tagging at the *sox10* locus and summary of our dual-cassette donor KI strategy. (**A**) Z-stack confocal images of 48 hpf zebrafish embryos after the injection of the *sox10* geno-tagging donor knockin system at the one-cell stage (Control) followed by the further injection of 25 pg *Cre* mRNA into a single cell at the 4 cell stage (+*Cre* mRNA). White arrowheads indicate the colocalization of the tdGFP and tdTomato signals. Scale bar, 100 μm. (**B**) Junction PCR and clonal sequencing to detect the knockin and *Cre*-induced recombination events in the injected embryos

*Figure 4 continued on next page*

*Figure 4 continued*

showing mosaic double-fluorescence signals. Injected: Embryos injected with the donor, Cas9/gRNA system and *Cre* mRNA. Donor: *sox10* geno-tagging donor plasmid. Uninjected: Uninjected embryos. (C) Summary of the applications and potential expansion of our dual-cassette donor KI strategy.

DOI: https://doi.org/10.7554/eLife.48081.010

The following figure supplement is available for figure 4:

**Figure supplement 1.** Generation and evaluation of the *sox10* geno-tagging allele.

DOI: https://doi.org/10.7554/eLife.48081.011

of the cells expressing the target gene, through the incorporation of an in-frame fluorescent reporter gene (*tdTomato* or *tdGFP*) in the positive-cassette of the donor (the feasibility of this dual-function 'two-in-one' KI allele strategy was demonstrated at zebrafish *tbx5a* and *kctd10* loci); (iii) differential labeling of two different alleles (normal *vs* defective) to achieve a geno-tagging effect via the differential labeling of the two cassettes (Po- *vs* Ne-cassette) in the donor with two different fluorescent reporters. This advanced strategy enables the conditional switching of the fluorescent reporters and, thus, the discrimination of cells with three different genotypes (homozygous normal, heterozygote, and homozygous mutant/defective). We successfully generated such conditional coupled geno-tagging alleles at the *tbx5a* and *sox10* loci in zebrafish and demonstrated the fluorescent reporter switch in response to Cre recombinase treatment. As the activity of Cre recombinase can easily be manipulated (*e.g.*, by using inducible CreER$^{T2}$), genetic mosaicism of target tissues could be generated through conditional knockout. However, previously reported KI techniques usually cannot distinguish cells or tissues with different genotypes, which limits the in-depth study of gene functions. With simple modifications of our dual-cassette donor system, we have established a novel geno-tagging technique that allows conditional knockout and geno-tagging switching to occur simultaneously in the same allele. Fluorescence label switching, together with conditional gene knockout, not only allows the real-time labeling and discrimination of knockout cells or tissues from normal ones to trace and compare their fates in vivo but can even go further, potentially labeling different alleles with different fluorescent reporters. In this manner, one can easily determine and follow the genotype of each allele and each cell.

In addition, there is another potential advantage of the geno-tagging strategy due to the special design of the Ne-cassette in the donor. While the original single-fluorescent-reporter version of the dual-cassette donor might trigger genetic compensation since it contains a *mutExon* bearing a premature stop codon in its Ne-cassette, the mutant alleles produced through the geno-tagging KI strategy, which does not contain a premature stop codon to disrupt the target gene, are likely exempted from the disturbance caused by the genetic compensation response, thus simplifying as well as consolidating subsequent phenotype analyses. Therefore, we recommend the use of the geno-tagging donor for the generation of CKO alleles coupled with gene or allele labeling effects. Regarding polyA signals, in principle, more polyA signals seem to be more effective for efficient termination of transcription (*Ni et al., 2012*). Although *2PA* was satisfactory for the generation of the *tbx5a*$^{PoR-NeG}$ geno-tagging allele in the present study, we adopted eight tandem repeats of SV40 polyA signals (*8xPA*) for stronger transcriptional termination for the *sox10* locus. However, there are potential disadvantages of employing two *8xPA* sequences, each containing eight tandem repeats of the ~160 bp polyA signal sequence, concerning stability. The repeats within each *8xPA* sequence might be reduced during expansion and/or passaging to the next generations of the *sox10*$^{PoR-NeG}$ allele; on the other hand, the sequence flanked by the two *8xPA* sequences (containing the downstream *loxP* and Ne-cassette) might be lost due to occasional recombination of the two *8xPA* sequences, leading to the abolition of the CKO and fluorescence-switch effects. Practically, since different target genes may have very different expression patterns and regulation properties, for each particular target gene, we recommend beginning with the construction of the donor with *2PA*, then testing its effect in founder embryos to determine whether this is sufficient to stop the expression of the second fluorescent reporter, and introducing more polyA signals when necessary.

Germline transmission is crucial for the successful establishment of stable KI transgenic fish lines. Interestingly, our results indicate a strong correlation between the efficient germline transmission and correct insertion of KI donors in somatic tissues. We observed high-efficiency germline recovery of the KI events after preselection for mosaic founders that acquired the correct donor insertions in

somatic tissues by selecting either embryos on the basis of fluorescent reporter signals or adult fish on the basis of fin clip junction PCR. During the construction of the *tbx5a* PoR-Ne KI fish line, approximately 15% of $F_0$ embryos showed the expected *tdTomato* expression after the injection of the donor together with the Cas9/gRNA system. More than 50% (5/9) of these fluorescent reporter-positive $F_0$ fish successfully transmitted the KI alleles to their progeny. In contrast, no germline transmission events were identified in the 40 founders negative for the somatic cell expression of *tdTomato*. Similarly, approximately half (10/21) of the embryonic stage preselected founders showed germline transmission of the *sox10* geno-tagging KI alleles. For the screening of *tbx5a* PoR-NeG geno-tagging KI fish, we preselected adult fish for somatic cell integration by junction PCR of fin clips and found that 6 out of 48 founders were positive for donor insertions in this somatic tissue. Among these founders, 2 out of 5 positive founders were also positive for germline transmission. Not surprisingly, we also screened for germline transmission in 20 founders negative by fin clip junction PCR and found that none of them transmitted a KI donor to their offspring. In summary, to achieve efficient recovery of germline-transmissible KI alleles, we strongly recommend the preselection of $F_0$ embryos according to fluorescent signals or adults on the basis of fin clip junction PCR and then screening only these founders in germline transmission analysis.

Compared with traditional KI strategies, which generally adopt donors containing a single functional cassette, our dual-cassette donor design provides more flexibility to achieve complex genome manipulation, enabling a variety of applications. In addition to coupling conditional knockout with gene-labeling or geno-tagging, this dual-cassette donor system could theoretically be expanded to other applications after further modification (*Figure 4C*), such as to simulate precise genome modifications, achieve conditional gene rescue or differentiate and analyze maternal, zygotic and paternal gene expression and activity. For example, the Ne-cassette can be engineered to contain a coding sequence with a precise mutation (such as a disease-causing point mutation) so that the deletion of the Po-cassette could conditionally simulate the precise mutation. On the other hand, conditional gene rescue can be achieved by swapping the Po-cassette and Ne-cassette. This strategy may also be utilized to study cell-autonomous and cell-non-autonomous gene functions. Furthermore, our dual-cassette donor knockin strategy could theoretically be customized for use in other model systems, such as mice and *Drosophila*, or adapted for medical research, including the establishment of disease models for drug screening and mechanism dissection or the tracing of cancer cell transformation and metastasis to provide new insights into carcinogenesis and cancer therapy.

Nevertheless, our current dual-functional donor KI strategy has certain limitations and is open to further improvement. To simplify donor design and construction and reduce the size of the donor plasmids as much as possible, we have omitted most intron sequences downstream of the Cas9/gRNA target sites as well as the endogenous 3'-UTRs, and mainly relied on the downstream coding sequence from the cDNA to rescue the correct expression of the target gene. This approach may alter the expression pattern and/or regulation of the target gene, so we suggest careful evaluation of the reporter gene expression and comparison of the result with the endogenous gene characteristics before performing further experiments, such as conditional knockout or lineage tracing. Of course, one can easily include introns and 3'-UTRs in the donor design when necessary. On the other hand, since only one form of the downstream intronless cDNA sequence is inserted into the final KI allele to complement the disruption of endogenous genes, this approach may not be applicable to genes with multiple alternative splicing isoforms or genes regulated by RNA splicing. However, this limitation may be avoided by including all the downstream introns in the Po-cassette of the donor. In addition, this limitation sometimes might be considered a unique advantage, through which KI alleles with different alternative splicing isoforms could be purposely designed and generated separately, facilitating specific and detailed investigation as well as the comparison of different isoforms. In addition to the disadvantages in the design of the rescue sequences in the Po-cassette, another concern is that all the current KI alleles retain the plasmid backbone sequence at the insertion site in the genome, which might induce DNA methylation and silencing of transcription, as previously reported (*Suzuki et al., 2016*). In the future, we will try to improve the donor by introducing another recombination system, such as Flp/*FRT*, *attB*/*attP*, or Dre/*Rox* to flank the unwanted backbone and then delete it after successful knock in *Lu et al. (2011)*; *Wong et al. (2011)*; *Park and Leach, 2013*; *Carney and Mosimann (2018)*, or by introducing two Cas/gRNA target sites in the donor, flanking the unwanted backbone to separate it from the functional cassettes during targeted insertion, or by using minicircle DNA to minimize the unwanted backbone sequence.

# Materials and methods

**Key resources table**

| Reagent type (species) or resource | Designation | Source or reference | Identifiers | Additional information |
|---|---|---|---|---|
| Gene (*Danio rerio*) | *tbx5a* | | Gene: 30071 | ENSDARG00000024894 |
| Gene (*Danio rerio*) | *kctd10* | | Gene: 406787 | ENSDARG00000017115 |
| Gene (*Danio rerio*) | *sox10* | | Gene: 140616 | ENSDARG00000077467 |
| Strain, strain background (*Danio rerio*) | Tübingen (TU) | Our lab | | A commonly used wild-type zebrafish strain |
| Strain, strain background (*Danio rerio*) | *Tg(cmlc2:EGFP)* | **Tong et al., 2014** | PMID: 24430697 | |
| Strain, strain background (*Danio rerio*) | *Tg(cmlc2: zCreER$^{T2}$-T2A-ECFP)* | This paper | | A transgenic zebrafish expressing zebrafish codon-optimized *CreERT2* and *ECFP* driven by the heart-specific *cmlc2* promoter |
| Recombinant DNA reagent | pMD18-T vector (plasmid) | TAKARA | Cat#: 6011 | |
| Recombinant DNA reagent | pMD19-T simple vector (plasmid) | TAKARA | Cat#: 3271 | |
| Recombinant DNA reagent | pminiTol2 (plasmid) | **Balciunas et al., 2006** | PMID: 17096595 | |
| Recombinant DNA reagent | pGH-T7-zCas9 (plasmid) | Our lab | PMID: 24480746 | **Liu et al., 2014** |
| Recombinant DNA reagent | pMD18T-tdGFP (plasmid) | Dr. Yulong Li | | |
| Recombinant DNA reagent | pUC19-scaffold (plasmid) | Dr. Jingwei Xiong | PMID: 23528705 | **Chang et al., 2013** |
| Recombinant DNA reagent | bait otx2 CreER$^{T2}$ (plasmid) | Dr. Michael Brand | PMID: 29435650 | **Kesavan et al., 2018** |
| Recombinant DNA reagent | pX-T7-Cre (plasmid) | Dr. Yao Zu | | |
| Commercial assay or kit | Gibson assembly | NEB | E5510S | |
| Commercial assay or kit | mMessage mMachine T7 kit | Ambion | AM1344 | |
| Chemical compound, drug | 4-HT (Hydroxyltamoxifen) | Sigma | H6278-10MG | |
| Chemical compound, drug | Tricaine (ethyl 3-aminobenzoate methanesulfonate salt) | Sigma | E10521 | |
| Software, algorithm | AxioVision Rel.4.8 | Zeiss | RRID: SCR_002677 | |
| Software, algorithm | ZEN 2009 | Zeiss | RRID: SCR_013672 | |
| Other | TRIzol reagent | Invitrogen | Cat#: 10296028 | |
| Other | 5x All-In-One RT MasterMix | abm | G485 | |
| Other | EvaGreen 2x qRT-PCR Mastermix | abm | Mastermix-S | |

## Zebrafish husbandry

All the zebrafish used in this study were raised in groups and maintained at 28.5°C in the zebrafish facility of Peking University with a 14 hr/10 hr light/dark cycle. The wild-type strain used was Tübingen (TU). All animal experiments were approved by the Institutional Animal Care and Use Committee (IACUC) of Peking University. The reference from IACUC of Peking University is LSC-ZhangB-2.

## Donor construction

To construct the PoNe donor for the *tbx5a* locus targeting the second intron (*i.e.*, the *tbx5a-T2A-tdTomato floxP 2PA-mutExon* donor) as an example, the *hEMX1* target site and the first *loxP* site were linked by a 111 bp DNA fragment cloned from the pMD18-T vector (TAKARA, 6011) to avoid the disruption of the *loxP* sequence due to NHEJ-mediated DSB repair induced by the cleavage of the *hEMX1* sequence, and this *hEMX1-linker-loxP*[1st] sequence was cloned into the pMD19-T simple vector (TAKARA, 3271). Then, the intron two sequence downstream of the *tbx5a I2* target site (including the splice acceptor) of the *tbx5a* gene was cloned downstream of this first *loxP* site. The *tbx5a* partial CDS (referred to as 'E3+Remaining CDS' in *Figure 1A*) (except for exon 1, exon two and the stop codon) was fused with the *T2A-tdTomato* sequence. Finally, the second *loxP* site, SV40 *PA* (polyA signal), BGH *PA* and a modified exon 3 of *tbx5a* that contained an in-frame stop codon and included its upstream splice acceptor sequence, were successively cloned downstream of the *tdTomato* coding sequence. The *tbx5a-T2A-tdTomato floxP 2PA-mutExon* donor (abbreviated as *tbx5a* PoR-Ne donor) was used as the basic vector to construct the PoNe donor for the *kctd10* gene and for the geno-tagging donors for the *tbx5a* and *sox10* loci. To construct the *tbx5a* geno-tagging PoR-NeG donor as an example, we improved the initial *tbx5a* PoR-Ne donor by cloning the sequence around the splice acceptor of the second intron of the *tbx5a* gene and fused it in-frame with a *tdGFP* coding sequence, which was separated by a 2A peptide sequence, and we subsequently inserted the whole sequence between the second *loxP* site and the *2PA*; then, another copy of *2PA* was inserted upstream of the second *loxP* site, and the *mutExon* was removed (*Figure 3A*).

## Preparation of the gRNAs, *zCas9* mRNA and *Cre* mRNA

The gRNAs were designed with the CasOT program (http://casot.cbi.pku.edu.cn/) (*Xiao et al., 2014*). Forward oligonucleotides containing a T7 promoter, gRNA target site and partial gRNA scaffold sequences were designed for gRNA template synthesis through PCR amplification by using the pUC19-scaffold as the template (*Chang et al., 2013*), together with a universal reverse primer (5'-AAAAAAAGCACCGACTCGGTGCCAC-3'). Then, gRNAs were synthesized by in vitro transcription with T7 RNA polymerase (TAKARA) and purified by LiCl or ethanol precipitation. The gRNA target sequences are shown in *Supplementary file 1*. The zebrafish codon-optimized Cas9 expression vector pGH-T7-zCas9 was linearized by XbaI and used as the template for generating *zCas9* mRNA through in vitro transcription using the mMessage mMachine T7 kit (Ambion) (*Liu et al., 2014*). Then, the *zCas9* mRNA was purified by LiCl precipitation. The Cre expression vector pX-T7-Cre was linearized with XbaI digestion and used as the template for synthesizing *Cre* mRNA through in vitro transcription by using the mMessage mMachine T7 kit (Ambion). Then, the *Cre* mRNA was purified by LiCl precipitation.

## Microinjection of zebrafish embryos

The *zCas9* mRNA and gRNAs were coinjected into one-cell-stage zebrafish embryos with or without donor plasmids. Each embryo was injected with 1–2 nL of a solution containing 700 ng/μL *zCas9* mRNA, 100 ng/μL gRNA and 15 ng/μL donor plasmid; 100 pg or 25 pg of *Cre* mRNA was injected into 1 cell stage or a single cell of 4 cell stage zebrafish embryos, respectively.

## Restriction endonuclease assay to evaluate the indel efficiency of the Cas9/gRNA system

For indel efficiency evaluation, one-cell stage zebrafish embryos were injected with the *zCas9* mRNA and the corresponding gRNA. To extract genomic DNA, at 24 hpf, fifteen injected embryos were divided into three groups and lysed with 50 mM NaOH solution (10 μL each embryo) at 95°C for 15 min, followed by neutralization with 1 M Tris-HCl (pH = 8.0, 1 μL of each embryo). Then, 1 μL of the

genomic DNA extract was used as the template to amplify the target site regions with the corresponding primers (*Supplementary file 7*). The PCR products were subsequently digested by using the corresponding restriction endonucleases and analyzed by agarose gel electrophoresis and Sanger sequencing after TA cloning.

## Junction PCR and sequencing

Genomic DNA was extracted with lysis buffer (10 mM Tris-HCl, pH 8.2, 200 mM NaCl, 5% SDS solution, 200 µg/mL proteinase K and 10 mM EDTA) from either individual or pools of 72 hpf zebrafish embryos and then used to PCR amplify the 5' and 3' junction fragments of target genes or the region flanking the *loxP* site using the appropriate primers (*Supplementary file 7*). The PCR products were either directly sent for sequencing ($F_1$ and $F_2$ embryos) or cloned into pMD18-T (TAKARA) for clonal sequencing ($F_0$ embryos).

## Generation of transgenic fish and chemical treatment

To construct the plasmid for the generation of *Tg(cmlc2:zCreER^{T2}-T2A-ECFP)* zebrafish, the zebrafish *cmlc2* (*myl7*) promoter sequence and the SV40 polyadenylation site were cloned and ligated into the pminiTol2 vector (*Balciunas et al., 2006*) by the Gibson assembly method according to the manufacturer's instructions (NEB). Then, the coding regions of both zebrafish codon-optimized *CreER^{T2}* from the *bait otx2 CreER^{T2}* (*Kesavan et al., 2018*) and *ECFP* were cloned and ligated into this *Tol2* vector *via* the Gibson assembly method. Approximately 35 pg of the plasmid and 80 pg of *Tol2* transposase mRNA were coinjected into one-cell-stage fertilized zebrafish eggs to generate $F_0$ transgenic fish. $F_0$ adults were screened for germline transmission of *zCreER^{T2}-T2A-ECFP* expression vector integration by the observation of correct ECFP fluorescent signals in the progeny. To activate CreER^{T2}, zebrafish embryos were treated with 5 µM 4-HT (Sigma) from 5 hpf to 48 hpf and the embryo medium was changed once at 24 hpf.

## Imaging and processing

For general imaging, zebrafish embryos were anesthetized with 0.02% tricaine (ethyl 3-aminobenzoate methanesulfonate salt, 1 M Tris-HCl, pH 9.0), then placed in 3% methylcellulose (Sigma), and imaged under a microscope (AXIO Imager Z1; Zeiss) equipped with AxioCam MRm (Zeiss). Confocal imaging was performed using an LSM 710 confocal microscope (Zeiss) with a 20X water immersion objective. Z-stack images were acquired with a 5- to 7 µm resolution. The images were processed with AxioVision Rel.4.8 and ZEN 2009 imaging software.

## qRT-PCR analysis

RNA was isolated using TRIzol reagent (Invitrogen) according to the standard protocol from 10 randomly picked 72 hpf sibling embryos with or without the injection of *Cre* mRNA. All the embryos were collected from crosses of the zebrafish carrying the corresponding homozygous KI alleles with either a wild-type or a heterozygote zebrafish. Embryos of different genotypes were distinguished and grouped by fluorescent signals and corresponding phenotypes. At least 500 ng of total RNA was employed for reverse transcription using the 5x All-In-One RT MasterMix (Applied Biological Materials). All reactions were performed in at least technical triplicates, and the results represent biological triplicates. Each primer pair was designed to span the KI donor integration site, and *gapdh* was used as the internal reference gene. qRT-PCR primer information is listed in *Supplementary file 7*. qRT-PCR experiments were performed in a Roche LightCycler 96 instrument or BioRad CFX96 using EvaGreen 2x qRT-PCR Mastermix (Applied Biological Materials) and a 500 nM final primer concentration. Fold changes were calculated using the $2^{-\Delta\Delta Ct}$ method, and $p<0.05$ was accepted as statistically significant.

## Acknowledgements

We thank Yulong Li for providing us with the *pMD18T-tdGFP* vector. We thank Jingwei Xiong for providing us with the *pUC19-scaffold* plasmid. We thank Yao Zu for providing us with the *pX-T7-Cre* vector. We thank Dr. Michael Brand for kindly providing the *bait otx2 CreER^{T2}* plasmid. We thank Christopher J Krueger for language editing, Yuying Gao and Yan Shen for lab management, and

Yingdi Jia and Jingliang Chen for zebrafish husbandry. We also thank Xiaochen Li and Xuemei Hao from the Core Facilities at the School of Life Sciences of Peking University in Beijing, China, for assistance with confocal microscopy. This work was partially supported by grants from the National Key Research and Development Program of China [2018YFA0801000, 2016YFA0100500]; the National Key Basic Research Program of China [2015CB942803]; the National Natural Science Foundation of China (NSFC) [31671500, 31871458 and 81371264]; and the PKU Qidong-SLS Innovation Fund.

## Additional information

### Funding

| Funder | Grant reference number | Author |
|---|---|---|
| National Key Research and Development Program of China | 2018YFA0801000 | Bo Zhang |
| National Key Research and Development Program of China | 2016YFA0100500 | Bo Zhang |
| National Key Basic Research Program of China | 2015CB942803 | Bo Zhang |
| National Natural Science Foundation of China | 31671500 | Bo Zhang |
| National Natural Science Foundation of China | 31871458 | Bo Zhang |
| National Natural Science Foundation of China | 81371264 | Bo Zhang |
| Peking University | Qidong-SLS Innovation Fund | Bo Zhang |

The funders had no role in study design, data collection and interpretation, or the decision to submit the work for publication.

### Author contributions

Wenyuan Li, Conceptualization, Data curation, Formal analysis, Investigation, Visualization, Methodology, Writing—original draft; Yage Zhang, Bingzhou Han, Data curation, Formal analysis, Validation, Investigation, Visualization, Methodology, Writing—review and editing; Lianyan Li, Muhang Li, Xiaochan Lu, Cheng Chen, Mengjia Lu, Yujie Zhang, Xuefeng Jia, Investigation; Zuoyan Zhu, Supervision; Xiangjun Tong, Conceptualization, Supervision; Bo Zhang, Conceptualization, Supervision, Funding acquisition, Project administration, Writing—review and editing

### Author ORCIDs

Yujie Zhang (iD) http://orcid.org/0000-0001-5038-1487
Bo Zhang (iD) https://orcid.org/0000-0002-6436-5629

### Ethics

Animal experimentation: All animal experiments were approved by Institutional Animal Care and Use Committee (IACUC) of Peking University. The reference from IACUC of Peking University is LSC-ZhangB-2.

### Decision letter and Author response

Decision letter https://doi.org/10.7554/eLife.48081.021
Author response https://doi.org/10.7554/eLife.48081.022

## Additional files

### Supplementary files

• Supplementary file 1. The Cas9/gRNA target sequences used in this study.
DOI: https://doi.org/10.7554/eLife.48081.012

• Supplementary file 2. Indel and knockin efficiencies in founder embryos.
DOI: https://doi.org/10.7554/eLife.48081.013

• Supplementary file 3. Germline mosaicism of the *tbx5a* PoR-Ne donor KI in each positive $F_0$.
DOI: https://doi.org/10.7554/eLife.48081.014

• Supplementary file 4. Germline mosaicism of the *kctd10* PoG-Ne donor KI in each positive $F_0$.
DOI: https://doi.org/10.7554/eLife.48081.015

• Supplementary file 5. Germline mosaicism of the *tbx5a* geno-tagging PoR-NeG donor KI in $F_0$.
DOI: https://doi.org/10.7554/eLife.48081.016

• Supplementary file 6. Germline mosaicism of the *sox10* geno-tagging PoR-NeG donor KI in $F_0$.
DOI: https://doi.org/10.7554/eLife.48081.017

• Supplementary file 7. The sequences of the primers used for PCR and qRT-PCR analyses.
DOI: https://doi.org/10.7554/eLife.48081.018

• Transparent reporting form  DOI: https://doi.org/10.7554/eLife.48081.019

### Data availability

All data generated or analysed during this study are included in the manuscript and supporting files.

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
