## [Decision Letter]

**Acceptance summary:**

Your manuscript represents a very important technical advance in the emerging field of conditional mutagenesis in the zebrafish. The mutagenesis vectors you have developed offer a key advantage over existing tools: the ability to track homozygous wild type, heterozygous mutant or homozygous mutant status of every cell using fluorescent reporters. Together with well-documented amenability of zebrafish embryos and larvae to high-resolution imaging, these tools should enable tremendous advances in genetic analysis of developmental processes. Furthermore, the ability to unequivocally assign mutant, heterozygous or wild type genotypes to every cell in a mosaic tissue should be of high interest to those studying genetic control of post-embryonic processes such as regeneration. The apparent ease with which your vectors can be integrated into the genome (using non-homologous end joining instead of less efficient but more precise homology directed repair) should make the methodology readily accessible to a large number of laboratories.

**Decision letter after peer review:**

Thank you for submitting your article "One-step efficient generation of dual-function conditional knockout and geno-tagging allele in zebrafish" for consideration by *eLife*. Your article has been reviewed by Didier Stainier as the Senior Editor, a Reviewing Editor, and three reviewers. The following individuals involved in review of your submission have agreed to reveal their identity: David Grunwald (Reviewer #3).

The reviewers have discussed the reviews with one another and the Reviewing Editor has drafted this decision to help you prepare a revised submission.

Summary:

The manuscript by Zhang and colleagues describes a novel conditional mutagenesis system for the zebrafish. The system uses non-homologous end joining pathway to integrate dual function transgenes into the genome with very high efficiency. The "rescue" component of the integrated transgene contains the cDNA coding for exons downstream of the integration site co-translationally linked to a fluorescent reporter. The "rescue" cassette is flanked by *loxP* sites for Cre-mediated excision. The second part of the integrated transgene contains transcriptional terminators and splice acceptors for mutagenesis using the gene trap principle. Authors demonstrate efficient transgene integration and Cre-dependent mutagenesis of *tbx5a, sox10* and *kcdt10* genes.

While several recent papers have described conditional mutagenesis methods for zebrafish, they have not yet been broadly employed by other laboratories. The approach described in this manuscript is made particularly attractive by the ability to pre-screen using fluorescence, resulting in very high apparent transgenesis rates. The ability to use fluorescence to track mutant and wild type alleles built into the PoR-NeG vectors may be of particular interest for mosaic analysis. An additional advantage, shared with another method recently published in *eLife* (Sugimoto et al., 2017), is that downstream exons of the mutated gene are not expressed, which should help avoid genetic compensation.

In summary, the manuscript by Zhang et al. represents an important step developing a highly sought-after technique of conditional mutagenesis for the zebrafish.

Essential revisions:

1) The described method of conditional mutagenesis is very likely to be refractive to genetic compensation (with the caveat that the *mutExon* in the PoNe donor may induce NMD and genetic compensation if expressed). This compelling hypothesis can only be tested by mutating a gene known to be subject to genetic compensation. To our knowledge, *tbx5a, sox10* and *kcdt10* loci selected for mutagenesis are not subject to genetic compensation, making them unsuitable for testing the non-compensation hypothesis. Thus, authors must either provide data demonstrating lack of genetic compensation or text sentences describing genetic compensation to the Discussion section, clearly indicating speculative nature of such statements. Furthermore, the possibility that *mutExon* in PoNe donor may induce compensation needs to be discussed.

2) Expression of insertionally mutated loci has to be quantified in "on" and "off" states. It can be done by qRT-PCR using primers in exons upstream and downstream of the integration site, as is common for gene traps. The need to analyze expression in the "on" state is justified because the gene is expressed from intron-less cDNA with a non-native 3' UTR. It is important to analyze expression in the "off" state because all the coding sequences are present in the genome and mutagenicity relies on the efficiency of splicing and transcriptional termination modules within the inserted transgene.

3) A thorough breakdown of transgenesis efficiency must be provided, starting with numbers of injected embryos (if available), embryos screened for fluorescence, percentage/numbers of embryos selected for raising and so forth.

4) A better explanation for different components used in different vectors must be provided. Specifically, why was *mutExon* not retained PoR-NeG vectors, why different transcriptional terminators are used in PoR-NeG vectors integrated into *tbx5a* and *sox10* loci? Stability of constructs and transgenes containing two 8X terminators should be discussed. In simplest terms, which vector would you recommend for other labs to use?

5) Since rescue function is provided by partially intronless cDNA, this approach may not be applicable to genes with multiple alternatively spliced isoforms or genes regulated at the level of RNA splicing. This limitation should be discussed.

6) Higher resolution images of transgenes must be included for all figures, especially Figure 4—figure supplement 1.

*Reviewer #1:*

The manuscript by Li et al., "One-step efficient generation of dual-function conditional knockout and geno-tagging allele in zebrafish", describes the authors successful attempts in generating dual functional alleles in zebrafish using NHEJ-mediated insertion targeted by Cas9/gRNA. This gene-trap approach uses two functional units in the vectors. The first unit confers cis-complementation of the mutagenic effect and expression of a fluorescent protein. It consists of the native splice acceptor, the remaining downstream coding sequence, and the coding sequence of a fluorescent protein. The unit is flanked by *loxP* sites to allow Cre-dependent removal. The second unit confers transcriptional termination. In the basic vector, it consists of two or more pA signals (SV40 pA and BGH pA) followed by a mutant native exon with a premature stop codon. In the more advanced version, the second unit consists of a traditional gene-trap that could lead to the expression of a different fluorescent protein. The authors inserted the cassettes at 3 different loci (*tbx5a, kctd10* and *sox10*) and demonstrated germline transmission in 5-50% founders. They showed the conditionality of these alleles and the utility of alleles generated by the advanced cassette in revealing the genotypes of individual cells. The results are of good quality and the writing is fair. Efficient generation of conditional alleles is a sought-after technique in the zebrafish field. The manuscript therefore addresses an unmet need and should be of interest to scientists in the zebrafish community and beyond. However, there are several issues that need to be addressed, both in the experiments and in the writing.

Essential revisions:

1) There is no quantitative evaluation of the effects of the KI alleles on gene expression. The authors only qualitatively assessed the KI alleles by comparing the expression patterns of the fluorescent protein(s) and the major phenotypes of the homozygous mutants. Quantitative evaluation of such alleles in "normal" and "defective" state at the mRNA level is a norm and should be done here. It is important to know how normal is "normal" and how defective is "defective".

2) The cassette design seems random, not systematic. The authors used 3 different designs. They discussed potential benefits of each feature but often without strong experimental evidence. When adding more features to the more advanced cassettes, they also dropped certain presumably important feature without rationale. For example, the authors argument that the mutant exon with premature termination codon in the cassette may be indispensable for effective gene disruption, but there is no direct evidence. They then dropped the mutant exon in the more advanced cassettes. Although the authors argued that *2PA* is highly effective as the transcriptional terminator, they replaced it with *8PA* in the last cassette. The stability of two tandem *8PA* needs to be evaluated.

3) The authors argued that one of the main advantages of the described alleles is that they may bypass transcriptional adaptation elicited by a premature termination codon. Yet they included a premature termination codon in their cassette.

4) All the alleles also retain the backbone sequence of the vector at the insert site. It has been reported that such extraneous sequence can induce DNA methylation and silencing. This needs to be discussed.

5) The resultant "normal" alleles have reduced intron number, which may alter the expression, or eliminate certain alternative spliced forms. This needs to be discussed as potential disadvantage.

6) The necessity of LiCl precipitation of gRNA is confusing. Although the observation is intriguing, but it is only true for the *emx1* gRNA. It is unclear whether the *lamGolden* gRNA also requires LiCl precipitation for activity.

*Reviewer #2:*

In the present report by Li et al. the authors describe a genetic method to generate conditional knock out alleles in zebrafish based on the Cre/Lox system and CRISPR/CAS9 mediated NHEJ. With this technique the authors generate specific alleles in few target loci (*tbx5a, kctd10* and *sox10*). In each case they developed reporter alleles that were tagged with fluorescent proteins and could be inactivated by Cre mRNA injection.

The use of NHEJ manipulate endogenous loci in zebrafish with high efficiency was previously reported in various papers. In particular a strategy to tag endogenous proteins at the C-terminus in zebrafish was previously reported by Li et al., 2015. The present manuscript is an improvement of the current methods that consists basically in the design of two *loxP* sites around the artificial tagged exon that is inserted by NHEJ.

The authors claim that their strategy will offer novel advantages including the avoidance of genetic compensation mechanisms and the possibility to generate conditional knock out alleles. Although I agree in principle with these statements these are not shown in the current paper.

In particular the authors should use their method to silence a known locus that is triggering genetic compensation when inactivated with the insertion of classical indel mutations (as described in the recent literature).

In addition, it is important to test their floxed alleles, crossing them with Cre expressing lines in specific tissues (for instance the heart primordium) to show that tissue and temporal control of the recombination event can be achieved.

Without these experiments the paper shows only a very limited advancement in designing clever targeting plasmid but fails to proof the real power of this approach.

*Reviewer #3:*

Li et al. introduce a new method for creating conditional knockout alleles in the zebrafish. The method generates a wildtype allele translationally linked to a fluorescent protein so that cells expressing the wildtype allele can be identified. Cre activity leads to excision of the wildtype-fluorescent reporter sequences and leads to expression of a different reporter protein so that the loss-of-function allele is tagged. It is a very nice idea and it will be very useful. The method is not perfect in that the wildtype activity is supplied by a cDNA sequence and the claim is that the mutant allele will not trigger transcription compensation, which is not tested and may or may not be true, but this method is a true advance in the field and will be implemented by others. The method deserves to be published.

Overall the manuscript is very well written. Nevertheless, the manuscript has a number of areas that require clarification. At present, a number of the experiments presented in the figures are difficult to interpret. The paper will make a much bigger impact if the reader has an easy time understanding the experiments and the results.

1) The authors do not correctly describe the efficiency rates of generating insertions or the efficiency with which germline insertions are made or recovered. This is a very important point and *must* be corrected. The authors claim throughout the paper that they have improved the efficiency of insertion or of generating insertions that will be transmissible through the germline. This is not correct. They have created a method that allows for the efficient recovery of germline transmissible modified alleles. They have made a wonderful advance but it is not the same as improving recombination rates. The authors have come up with a wonderful method for pre-selecting embryos that have acquired insertions in somatic tissue, because integration in the correct orientation leads to expression of a fluorescent reporter under control of the targeted gene. A low percentage of injected embryos grow into normal-looking embryos with reporter expression (subsection “High efficient generation of a dual-function KI allele at zebrafish *tbx5a* locus” claims 10% in one experiment). Then among those animals pre-selected for the presence integration events, a reasonable and varied fraction will transmit the edited allele to the next generation. Therefore, it is simply false for the authors to write "The feasibility of this strategy was demonstrated at *tbx5a* and *kctd1* loci, with germline transmission efficiency as high as 56%." Such a statement would be interpreted by readers as indicating that they had greatly advanced targeting efficiency so that 56% of injected animals will transmit an edited allele. Similarly, in the Discussion, the authors claim: "As high as 50% of the F_0_ fish could transmit the integrated donor construct to their offspring, which is much higher in germline transmission efficiency than previously reported HR-mediated gene knock-in." Instead, the authors should say they have developed a clever method that makes recovery of conditional alleles very efficient. This method involves first preselecting embryos that have mosaically acquired the conditional allele, and then only screening the preselected transgenic animals for the ability to transmit the alleles through the germline. Under these conditions, up to 56% of the pre-selected founders may transmit edited alleles. The authors should make this clear in the Abstract also.

2) It is difficult to reconstruct and to figure out the actual numbers that describe the recovery of germline transmissible alleles. For example, let's look at the section that begins subsection "Generation and evaluation of geno-tagging alleles at zebrafish *tbx5a* locus". The authors write: "After injection and screening, integration of this donor and germline transmission of the *tbx5a* PoR-NeG donor geno-tagging alleles were detected and confirmed in 2 out of 48 adult F_0_". I can't understand where the number "48" comes from – was this all the F_0_ adults from an injection? Was it 48 F_0_ selected on the basis of fluorescent reporter expression? In Table 2, row 3 it appears that 16/124 injected embryos had reporter expression – so where did the 48 come from?

3) The authors create conditional alleles that can be switched from WT red to mutant green. They call these PoR-NeG alleles for Positive Red to Negative Green. They also seem to call these geno-tagged alleles. They sometimes breed these to animals with another type of condition allele that switches from a WT-fluorescent reporter to a mutant-no reporter allele. They call these alleles PoNe, positive to negative. It would make the reading much easier if they always labeled an allele that co-expressed a fluorescent reporter. In other words, when the wildtype allele is marked by expression of tdTomato and the mutant allele is not marked by reporter expression, maybe call it PoR-Ne. In addition, they place these PoNe alleles on backgrounds that express a heart reporter (*cmcl2:EGFP*). As a result, if the WT allele were clearly marked as PoR-Ne then it would be easy to tell which is the heart marker and which is the tagged gene. In general: please use consistent nomenclature.

4) The figures need some work. The easiest suggestion is that the lettering needs to be larger or at higher resolution – especially for the line drawings – when I expanded it on my screen it became very pixelated. For example, try enlarging Figure 4—figure supplement 1F.

5) Figure 2A and C: I find these very hard to interpret – I think the authors are visualizing fluorescent markers in three embryo siblings generated from a single cross some of which have been injected with Cre mRNA. The patterns of fluorescence reporter expression differ and I believe the results are to be interpreted as being derived from embryos with different genotypes. Would it be possible for the authors to tell us the presumed genotypes of each row before Cre exposure? I feel this would make the interpretation easier for the reader. I don't understand Figure 2C – it looks like each embryo has a green heart but I think the authors want us to focus on differences. These need to be spelled out perhaps in the figure legend? For Panel E please explain in the figure legend this experiment. Explain the cross and what Normal and Defective embryos are. Explain what "before injection" means.

6) Figure 3B: I suspect the markers on the gel are mistakenly labeled. For example, the experimental band of 380 bp is larger than the marker of 500 bp and the experimental band of 820 bp is smaller than the marker of 750 bp. Also, in Panel B, the gel of the 3' junctions: I don't understand how amplification from the donor shown in Panel A with T5F2 and T5R1 would give a band – is there some mistake here?

7) In Figure 3C, I believe the images of dorsal view and ventral view have been reversed. I think the first column and bottom row picture is a ventral view. Please check – I may be wrong. I think the cross that produced the embryos in Panel C needs to be clearly explained and how to interpret the genotypes should be clearly explained. The images are difficult to interpret – what are we supposed to look at in the boxed insets? Explain in the legend please.

8) The images in Figure 3—figure supplement 1 are difficult to interpret. Again I suspect that if the authors told us the predicted genotypes and told us what we are supposed to be seeing in the outlined boxed areas, then it would be clearer.

9) Subsection “Generation and evaluation of dual-function alleles showing CKO with gene labeling effect at zebrafish *kctd10* locus”: it the sentence beginning “Interestingly, in the initial design of the *kctd10* PoNe donor…” might be easier to read if you re-phrased this sentence something like: "In initial experiments, we used only a donor with a single SV40pA termination signal sequence without the modified exon (*mutExon*). With such donors we were able to isolate stable integrations at the *kctd10* locus and observed the correct expression pattern of *tdGFP*.”

[Editors' note: further revisions were requested prior to acceptance, as described below.]

Thank you for resubmitting your work entitled "One-step efficient generation of dual-function conditional knockout and geno-tagging allele in zebrafish" for further consideration at *eLife*. Your revised article has been favorably evaluated by Didier Stainier (Senior Editor) and a Reviewing Editor.

The manuscript has been improved but there are some remaining issues that need to be addressed before acceptance, as outlined below:

1) The revised manuscript is much improved, and additional data demonstrating feasibility of fully conditional knockout using the new cmlc2:CreERT2 line. However, one major issue remains. It is common practice to measure the expression of targeted loci in the "off" state in homozygotes. Data can be displayed in several different ways, exemplified by Figure 2 in Ni et al., 2012 reference and Figure 1D in the Grajevskaja et al., 2018. Figure 1A in Sugimoto et al., 2017 serves as a directly relevant example of from a recently published *eLife* paper describing a conditional knock-in allele. This is very important. As one can appreciate from error bars in Figure 2—figure supplement 1E, Figure 2—figure supplement 2L, Figure 3—figure supplement 1E and Figure 4—figure supplement 1G, it is impossible to distinguish between 90% and 99% mutagenesis efficiency in heterozygotes (55% vs. 51% of read-through transcript remaining in heterozygotes). In contrast, the difference between 10% and 1% levels of remaining wild-type transcript should be quite straightforward to reliably quantify in homozygotes. Results of such quantitative analysis may have an impact on the choice between your described polyA cassettes (tandem SV40/GBH vs. 8xSV40). Thus, qRT-PCR analyses must be performed on embryos homozygous for Cre-excised alleles.

2) A second issue somewhat detracting from the overall quality of the work is occasionally difficult-to-read English. I would simply recommend that the final submission should be edited with the help of a professional editor.

3) Sequences of primers used for qRT-PCR should be included, along with other primers, in Supplementary file 7.

4) Why is *kctd10* deleted from the Abstract?

5) Subsection “High efficient generation of a dual-function KI allele at zebrafish *tbx5a* locus”: Need to show *hEMX1* target site on the vector in Figure 1A, and corresponding sites in vector diagrams in subsequent figures.

---

## [Author Response]

Essential revisions:1) The described method of conditional mutagenesis is very likely to be refractive to genetic compensation (with the caveat that the mutExon in the PoNe donor may induce NMD and genetic compensation if expressed). This compelling hypothesis can only be tested by mutating a gene known to be subject to genetic compensation. To our knowledge, tbx5a, sox10 and kcdt10 loci selected for mutagenesis are not subject to genetic compensation, making them unsuitable for testing the non-compensation hypothesis. Thus, authors must either provide data demonstrating lack of genetic compensation or text sentences describing genetic compensation to the Discussion section, clearly indicating speculative nature of such statements. Furthermore, the possibility that mutExon in PoNe donor may induce compensation needs to be discussed.

Sorry for the confusion in our statements about genetic compensation. We have tried to generate CKO+gene-labeling and geno-tagging alleles for the *egfl7* locus, to evaluate the possibility of eliminating genetic compensation by our method. Unfortunately, we have not identified satisfactory high-efficient Cas9/gRNA target sites in the introns of this gene so far, preventing us from generating KI lines for the evaluation. We have modified our statements about genetic compensation following your suggestions, including indicating speculative nature of the non-compensation claims as well as the possibility that *mutExon* in the PoNe donor may induce compensation, and moved the corresponding sentences to Discussion section.

2) Expression of insertionally mutated loci has to be quantified in "on" and "off" states. It can be done by qRT-PCR using primers in exons upstream and downstream of the integration site, as is common for gene traps. The need to analyze expression in the "on" state is justified because the gene is expressed from intron-less cDNA with a non-native 3' UTR. It is important to analyze expression in the "off" state because all the coding sequences are present in the genome and mutagenicity relies on the efficiency of splicing and transcriptional termination modules within the inserted transgene.

Thank you for the suggestion. We have designed primers flanking the insertion sites and performed qRT-PCR experiments to quantify the “on” and “off” states of the PoNe and geno-tagging alleles of *tbx5a, kctd10* and *sox10* loci, in 72 hpf embryos obtained from outcross of respective KI homozygous fish with a wild-type one, with or without *Cre* mRNA injection. As an example (subsection “Conditional knock out could be achieved for the *tbx5a* KI allele”), the results from the *tbx5a* PoR-Ne donor KI fish showed that the mRNA transcription level of the *tbx5a* locus in the *tbx5a^+/PoR-Ne^* embryos before *Cre* mRNA injection was comparable with that of wild-type control embryos, indicating normal expression of the *tbx5a^PoR-Ne^* allele (Figure 2—figure supplement 1E). After *Cre* mRNA injection, no red fluorescence signal could be detected in the *tbx5a^+/PoR-Ne^* embryos, indicating efficient excision of the Po-cassette and most of the *tbx5a^PoR-Ne^* alleles had been efficiently converted into *tbx5a^Ne^* alleles. The *tbx5a* expression level in these *tbx5a^+/Ne^* embryos significantly decreased to around 50% of the level in the wild-type embryos, as expected (Figure 2—figure supplement 1E). Similar results were obtained for the cross from either *tbx5a* or *sox10* PoR-NeG geno-tagging fish (subsection “Generation and evaluation of geno-tagging alleles at zebrafish *tbx5a* locus” and subsection “Geno-tagging effect achieved for zebrafish *sox10* gene”; Figure 3—figure supplement 1E, Figure 4—figure supplement 1G), or *kctd10* PoG-Ne KI fish (subsection “Generation and evaluation of dual-function alleles showing CKO with gene labeling effect at zebrafish *kctd10* locus”; Figure 2—figure supplement 2L).

3) A thorough breakdown of transgenesis efficiency must be provided, starting with numbers of injected embryos (if available), embryos screened for fluorescence, percentage/numbers of embryos selected for raising and so forth.

Sorry for the incompleteness of these important details about the injection and screening process. We have provided more detailed information about these process in the revised manuscript, as the following:

1) An extract from subsection “Germline transmission and evaluation of the *tbx5a* KI allele”: “To generate *tbx5a* PoR-Ne KI zebrafish, about 100 one-cell stage embryos were injected, and red fluorescent signals could be detected in 18 of them (Supplementary file 2). […] In contrast, no red fluorescent signal was detected in any F_1_ progeny from outcross of 40 adult F_0_ raised from tdTomato-negative embryos, indicating a positive correlation of reporter gene expression in founder embryos with successful as well as efficient germline transmission of the KI events.”

2) An extract from subsection “Generation and evaluation of dual-function alleles showing CKO with gene labeling effect at zebrafish *kctd10* locus”: “For the generation of *kctd10* PoG-Ne KI fish line, we injected about 100 one-cell stage embryos and observed the mosaic expression of tdGFP in 22 F_0_ embryos (Supplementary file 2). We raised all the F_0_ embryos without pre-selection to adulthood, and found 2 of 42 (4.8%) displayed germline transmission screening of 42 F_0_ adults, only two of them displayed germline transmission of the *kctd10* PoG-Ne donor KI events (Supplementary file 4).”

3) An extract from subsection “Generation and evaluation of geno-tagging alleles at zebrafish *tbx5a* locus”: “To generate *tbx5a* PoR-NeG geno-tagging zebrafish, about 100 one-cell stage embryos were injected, and 16 of them showed mosaic expression of red fluorescent signals (Supplementary file 2), whose ratio is comparable with the previous *tbx5a* PoR-Ne donor KI experiment. […] These results indicate a strong correlation between germline transmission of KI events and adult somatic tissue integration of the donor, and provide another efficient method for the enrichment of germline transmission events.”

4) An extract from subsection “Geno-tagging effect achieved for zebrafish *sox10* gene”: “To generate the heritable *sox10* PoR-NeG geno-tagging allele, about 150 one-cell stage embryos were injected with the KI system (without *Cre* mRNA), and the tdTomato fluorescent signals were observed in 103 F_0_ embryos (Supplementary file 2). The positive founder embryos showing fluorescent signals were raised to adulthood and germline transmission screening was conducted by outcross with wild-type zebrafish. Correct red fluorescent patterns in F_1_ embryos were observed in 10 out of 21 founders (Supplementary file 6).”

In addition, we added a new paragraph in the Discussion section and summarized as well as further discussed the striking correlation between efficient germline transmission and correct insertions of the KI donors in somatic tissues.

4) A better explanation for different components used in different vectors must be provided. Specifically, why was mutExon not retained PoR-NeG vectors, why different transcriptional terminators are used in PoR-NeG vectors integrated into tbx5a and sox10 loci? Stability of constructs and transgenes containing two 8X terminators should be discussed. In simplest terms, which vector would you recommend for other labs to use?

Sorry for the confusion. The key to the success of our dual-function KI strategy relies on the special design of the dual-cassette donor. In order to refine our method, we have tried slightly different designs on certain components in different KI experiments, on the premise of keeping the Po- and Ne-cassettes intact and functional. Since polyA signals are generally not 100% efficient for the termination of transcription, a *mutExon* (including its preceding splice acceptor sequence) bearing a premature stop codon was included in the Ne-cassette, following the 2X *PA* terminators, of the initial version of the PoNe donor, where only a single fluorescent reporter gene (*tdTomato* or *tdGFP*) was incorporated into the Po-cassette. In case there is leaky transcription after the 2X *PA*, this *mutExon* will be transcribed and spliced into the mature mRNA, ensuring disruption of the function of the target gene. This type of donor was used to generate the *tbx5a^PoR-Ne^, kctd10^PoG-Ne-1^* and *kctd10^PoG-Ne-2^* alleles. However, since the presence of *mutExon* will disrupt the translation of the fluorescent reporter in the Ne-cassette (of the PoR-NeG donor), and on the other hand, it might trigger genetic compensation, in the design of geno-tagging PoR-NeG donor, the advanced version of the original PoNe donor, we replaced the *mutExon* with its wild-type counterpart, to ensure normal expression of the fluorescent reporter in the Ne-cassette after Cre-induced removal of the Po-cassette, as well as to bypass the unwanted genetic compensation response.

In addition to avoiding the *mutExon* for the design of PoR-NeG geno-tagging donor, another concern is the efficiency of transcriptional termination of the Po-cassette, since its leaky transcription might lead to unwanted expression of the fluorescent reporter in the Ne-cassette before Cre-induction. It has been reported that the efficiency of transcriptional termination correlates with the number of polyA signals (Ni et al., 2012). Although we have obtained satisfactory results with the original 2X *PA* for the generation of *tbx5a^PoR-NeG^* geno-tagging allele, we decided to try more polyA signals for the *sox10* locus, concerning that the expression of *sox10* is more ubiquitous and seems also stronger than that of *tbx5a*. Therefore, to be on the safe side, we expanded the original single SV40 polyA signal into eight tandem repeats (*8xPA*) for stronger transcriptional termination in the Po-cassette, and also used this 8X *PA* sequence for the transcriptional termination of the Ne-cassette. Indeed, there are potential disadvantages for the employment of two 8X *PA* sequences, each containing eight times tandem repeats of the ~160-bp polyA signal sequence, in the *sox10* PoR-NeG geno-tagging donor as well as the *sox10^PoR-NeG^* allele, when concerning the stability. On one hand, the repeats within each 8X *PA* might be reduced during expansions and/or passages to next generations of this allele; On the other hand, the sequence flanked by the two 8X *PA* (containing the downstream *loxP* and the Ne-cassette) might be lost due to occasional recombination of the two 8X PA, leading to abolishment of the CKO and fluorescence-switch effect. Unfortunately, at present we have’t been able to assess the situation of these 8X *PA* sequences in the transgenic fish by PCR amplification and sequencing, due to its highly repetitive nature. However, so far, the *sox10^PoR-NeG^* allele behaves normally, considering the observation that the F_2_ embryos displayed expected expression of *tdTomato* signal under normal condtions and also showed expected switch to *tdGFP* expression upon treatment with Cre recombinase, indicating that the two 8X *PA* sequences are still functional. In the future, we will keep tracking the inheritance and stability of this allele, and also will try to establish a method to evaluate the situation of the 8X *PA* sequences in the transgenic fish.

Taking into account of all the above concerns for the donor design, we recommend to use the geno-tagging donor for the generation of CKO allele coupled with gene or allele labeling effect, beginning with constructing the donor with 2X *PA*, and then testing its effect in founder embryos to see whether it is enough to stop the expression of the second fluorescent reporter, and introducing more polyA signals when necessary.

We have clarified this issue by including the above explanations and discussions in the sections of both Results section and Discussion section in our revised manuscript.

5) Since rescue function is provided by partially intronless cDNA, this approach may not be applicable to genes with multiple alternatively spliced isoforms or genes regulated at the level of RNA splicing. This limitation should be discussed.

Thank you for reminding us the limitations of our method. We have clarified this issue by adding a new paragraph in the Discussion section, discussed the limitations of our current strategy, including the concerns about splicing regulation, and also gave some suggestions of potential solutions to these problems.

6) Higher resolution images of transgenes must be included for all figures, especially Figure 4—figure supplement 1.

Sorry for the quality of the figures in the manuscript of our initial submission. Now we have found a better method to preserve high resolution of our figures during conversion into the PDF file, and we believe that the new figures are satisfactory in quality and could display clear details after enlargement.

Reviewer #1:1) There is no quantitative evaluation of the effects of the KI alleles on gene expression. The authors only qualitatively assessed the KI alleles by comparing the expression patterns of the fluorescent protein(s) and the major phenotypes of the homozygous mutants. Quantitative evaluation of such alleles in "normal" and "defective" state at the mRNA level is a norm and should be done here. It is important to know how normal is "normal" and how defective is "defective".

Thank you for the suggestion. We have designed primers flanking the insertion sites and performed qRT-PCR experiments to quantify the “normal” and “defective” states of the PoNe and geno-tagging alleles of *tbx5a, kctd10* and *sox10* loci, in 72 hpf embryos obtained from outcross of respective KI homozygous fish with a wild-type one, with or without *Cre* mRNA injection. As an example (subsection “Conditional knock out could be achieved for the *tbx5a* KI allele”), the results from the *tbx5a* PoR-Ne donor KI fish showed that the mRNA transcription level of the *tbx5a* locus in the *tbx5a^+/PoR-Ne^* embryos before *Cre* mRNA injection was comparable with that of wild-type control embryos, indicating normal expression of the *tbx5a^PoR-Ne^* allele (Figure 2—figure supplement 1E). After *Cre* mRNA injection, no red fluorescence signal could be detected in the *tbx5a^+/PoR-Ne^* embryos, indicating efficient excision of the Po-cassette and most of the *tbx5a^PoR-Ne^* alleles had been efficiently converted into *tbx5a^Ne^* alleles. The *tbx5a* expression level in these *tbx5a^+/Ne^* embryos significantly decreased to around 50% of the level in the wild-type embryos, as expected (Figure 2—figure supplement 1E). Similar results were obtained for the cross from either *tbx5a* or *sox10* PoR-NeG geno-tagging fish (subsection “Generation and evaluation of geno-tagging alleles at zebrafish *tbx5a* locus”, subsection “Geno-tagging effect achieved for zebrafish *sox10* gene”; Figure 3—figure supplement 1E, Figure 4—figure supplement 1G), or *kctd10* PoG-Ne KI fish (subsection “Generation and evaluation of dual-function alleles showing CKO with gene 246 labeling effect at zebrafish *kctd10* locus”; Figure 2—figure supplement 2L).

2) The cassette design seems random, not systematic. The authors used 3 different designs. They discussed potential benefits of each feature but often without strong experimental evidence. When adding more features to the more advanced cassettes, they also dropped certain presumably important feature without rationale. For example, the authors argument that the mutant exon with premature termination codon in the cassette may be indispensable for effective gene disruption, but there is no direct evidence. They then dropped the mutant exon in the more advanced cassettes. Although the authors argued that 2PA is highly effective as the transcriptional terminator, they replaced it with 8PA in the last cassette. The stability of two tandem 8PA needs to be evaluated.

Sorry for the confusion. The key to the success of our dual-function KI strategy relies on the special design of the dual-cassette donor. In order to refine our method, we have tried slightly different designs on certain components in different KI experiments, on the premise of keeping the Po- and Ne-cassettes intact and functional. Since polyA signals are generally not 100% efficient for the termination of transcription, a *mutExon* (including its preceding splice acceptor sequence) bearing a premature stop codon was included in the Ne-cassette, following the 2X *PA* terminators, of the initial version of the PoNe donor, where only a single fluorescent reporter gene (*tdTomato* or *tdGFP*) was incorporated into the Po-cassette. In case there is leaky transcription after the 2X *PA*, this *mutExon* will be transcribed and spliced into the mature mRNA, ensuring disruption of the function of the target gene. This type of donor was used to generate the *tbx5a^PoR-Ne^, kctd10^PoG-Ne-1^* and *kctd10^PoG-Ne-2^* alleles. However, since the presence of *mutExon* will disrupt the translation of the fluorescent reporter in the Ne-cassette (of the PoR-NeG donor), and on the other hand, it might trigger genetic compensation, in the design of geno-tagging PoR-NeG donor, the advanced version of the original PoNe donor, we replaced the *mutExon* with its wild-type counterpart, to ensure normal expression of the fluorescent reporter in the Ne-cassette after Cre-induced removal of the Po-cassette, as well as to bypass the unwanted genetic compensation response.

In addition to avoiding the *mutExon* for the design of PoR-NeG geno-tagging donor, another concern is the efficiency of transcriptional termination of the Po-cassette, since its leaky transcription might lead to unwanted expression of the fluorescent reporter in the Ne-cassette before Cre-induction. It has been reported that the efficiency of transcriptional termination correlates with the number of polyA signals (Ni et al., 2012). Although we have obtained satisfactory results with the original 2X *PA* for the generation of *tbx5a^PoR-NeG^* geno-tagging allele, we decided to try more polyA signals for the *sox10* locus, concerning that the expression of *sox10* is more ubiquitous and seems also stronger than that of *tbx5a*. Therefore, to be on the safe side, we expanded the original single SV40 polyA signal into eight tandem repeats (*8xPA*) for stronger transcriptional termination in the Po-cassette, and also used this 8X *PA* sequence for the transcriptional termination of the Ne-cassette. Indeed, there are potential disadvantages for the employment of two 8X *PA* sequences, each containing eight times tandem repeats of the ~160-bp polyA signal sequence, in the *sox10* PoR-NeG geno-tagging donor as well as the *sox10^PoR-NeG^* allele, when concerning the stability. On one hand, the repeats within each 8X *PA* might be reduced during expansions and/or passages to next generations of this allele; On the other hand, the sequence flanked by the two 8X *PA* (containing the downstream *loxP* and the Ne-cassette) might be lost due to occasional recombination of the two 8X *PA*, leading to abolishment of the CKO and fluorescence-switch effect. Unfortunately, at present we have’t been able to assess the situation of these 8X *PA* sequences in the transgenic fish by PCR amplification and sequencing, due to its highly repetitive nature. However, so far the *sox10^PoR-NeG^* allele behaves normally, considering the observation that the F_2_ embryos displayed expected expression of *tdTomato* signal under normal condtions and also showed expected switch to *tdGFP* expression upon treatment with Cre recombinase, indicating that the two 8X *PA* sequences are still functional. In the future, we will keep tracking the inheritance and stability of this allele, and also will try to establish a method to evaluate the situation of the 8X *PA* sequences in the transgenic fish.

Taking into account of all the above concerns for the donor design, we recommend to use the geno-tagging donor for the generation of CKO allele coupled with gene or allele labeling effect, beginning with constructing the donor with 2X *PA*, and then testing its effect in founder embryos to see whether it is enough to stop the expression of the second fluorescent reporter, and introducing more polyA signals when necessary.

We have clarified this issue by including the above explanations and discussions in the sections of both Results section and Discussion section in our revised manuscript.

3) The authors argued that one of the main advantages of the described alleles is that they may bypass transcriptional adaptation elicited by a premature termination codon. Yet they included a premature termination codon in their cassette.

Sorry for the confusion. Yes, we indeed included a *mutExon* bearing a premature stop codon in the Ne-cassette of the original single fluorescent reporter version of the PoNe donor, as seen in the *tbx5a^PoR-Ne^, kctd10^PoG-Ne-1^* and *kctd10^PoG-Ne-2^* alleles, which are still prone to genetic compensation or transcriptional adaptation. This is one of the reasons for us to improve the design and came up with the advanced geno-tagging PoR-NeG donor, as represented by the *tbx5a^PoR-NeG^* and *sox10^PoR-NeG^* alleles, where the *mutExon* was replaced by its wild-type counterpart and another fluorescent reporter, which theoretically could be exempted from transcriptional adaptation. We have clarified this issue by adding the corresponding descriptions and discussions in our revised manuscript (subsection “Improving the dual-cassette-donor strategy to achieve geno-tagging effect”, Discussion section).

4) All the alleles also retain the backbone sequence of the vector at the insert site. It has been reported that such extraneous sequence can induce DNA methylation and silencing. This needs to be discussed.

Thank you for pointing out this issue. We have discussed this concern and also provided potential solutions in our revised manuscript (Discussion section).

5) The resultant "normal" alleles have reduced intron number, which may alter the expression, or eliminate certain alternative spliced forms. This needs to be discussed as potential disadvantage.

Thank you for pointing out this potential disadvantage. We have clarified this issue by adding a new paragraph in the Discussion, discussed the limitations of our current strategy, including the concerns about potential alteration of gene expression as well as splicing regulation, and also gave some suggestions of potential solutions to these problems (subsection “Geno-tagging effect achieved for zebrafish *sox10* gene”).

6) The necessity of LiCl precipitation of gRNA is confusing. Although the observation is intriguing, but it is only true for the emx1 gRNA. It is unclear whether the lamGolden gRNA also requires LiCl precipitation for activity.

Sorry for the confusion. We have compared the effect of LiCl precipitation with ethanol for the *lamGolden* gRNA and found similar result as *hEMX1* gRNA, where only LiCl precipitation gave satisfactory KI efficiency. We have clarified this issue by adding this information in our revised manuscript (subsection “Geno-tagging effect achieved for zebrafish *sox10* gene”).

Reviewer #2:In the present report by Li et al. the authors describe a genetic method to generate conditional knock out alleles in zebrafish based on the Cre/Lox system and CRISPR/CAS9 mediated NHEJ. With this technique the authors generate specific alleles in few target loci (tbx5a, kctd10 and sox10). In each case they developed reporter alleles that were tagged with fluorescent proteins and could be inactivated by Cre mRNA injection.The use of NHEJ manipulate endogenous loci in zebrafish with high efficiency was previously reported in various papers. In particular a strategy to tag endogenous proteins at the C-terminus in zebrafish was previously reported by Li et al., 2015. The present manuscript is an improvement of the current methods that consists basically in the design of two loxP sites around the artificial tagged exon that is inserted by NHEJ.The authors claim that their strategy will offer novel advantages including the avoidance of genetic compensation mechanisms and the possibility to generate conditional knock out alleles. Although I agree in principle with these statements these are not shown in the current paper.In particular the authors should use their method to silence a known locus that is triggering genetic compensation when inactivated with the insertion of classical indel mutations (as described in the recent literature).In addition, it is important to test their floxed alleles, crossing them with Cre expressing lines in specific tissues (for instance the heart primordium) to show that tissue and temporal control of the recombination event can be achieved.Without these experiments the paper shows only a very limited advancement in designing clever targeting plasmid but fails to proof the real power of this approach.

Thank you for the suggestions. We have tried to generate CKO+gene-labeling and geno-tagging alleles for the *egfl7* locus, to evaluate the possibility of eliminating genetic compensation by our method. Unfortunately, we have not identified satisfactory high-efficient Cas9/gRNA target sites in the introns of this gene so far, preventing us from generating KI lines for the evaluation. We have modified our statements about genetic compensation, indicating speculative nature of the non-compensation claims for our method, and moved the corresponding sentences to Discussion section.

In order to prove that tissue and temporal control of Cre-induced recombination events can be achieved in our KI transgenic zebrafish, we generated a heart specific *Cre* transgenic line *Tg(cmlc2:zCreERT2-2A-ECFP)* showing expression of both ECFP and 4-hydroxyltamoxifen (4-HT)-inducible CreER^T2^. This line was crossed with the *tbx5a^PoR-NeG/PoR-NeG^* geno-tagging adult fish, and the embryos were treated with 4-HT from 5 hpf to 48 hpf. As expected, green fluorescent signals were clearly detected in the ECFP-positive embryos but not in the un-treated control group, indicating successful switching of tdTomato to tdGFP expression (Figure 3—figure supplement 1D). We have included this new result in our revised manuscript (subsection “Generation and evaluation of geno-tagging alleles at zebrafish *tbx5a* locus”).

Reviewer #3:Overall the manuscript is very well written. Nevertheless, the manuscript has a number of areas that require clarification. At present, a number of the experiments presented in the figures are difficult to interpret. The paper will make a much bigger impact if the reader has an easy time understanding the experiments and the results.

*1) The authors do not correctly describe the efficiency rates of generating insertions or the efficiency with which germline insertions are made or recovered. This is a very important point and* must *be corrected. The authors claim throughout the paper that they have improved the efficiency of insertion or of generating insertions that will be transmissible through the germline. This is not correct. They have created a method that allows for the efficient recovery of germline transmissible modified alleles. They have made a wonderful advance but it is not the same as improving recombination rates. The authors have come up with a wonderful method for pre-selecting embryos that have acquired insertions in somatic tissue, because integration in the correct orientation leads to expression of a fluorescent reporter under control of the targeted gene. A low percentage of injected embryos grow into normal-looking embryos with reporter expression (subsection “High efficient generation of a dual-function KI allele at zebrafish tbx5a locus” claims 10% in one experiment). Then among those animals pre-selected for the presence integration events, a reasonable and varied fraction will transmit the edited allele to the next generation. Therefore, it is simply false for the authors to write "The feasibility of this strategy was demonstrated at tbx5a and kctd1 loci, with germline transmission efficiency as high as 56%." Such a statement would be interpreted by readers as indicating that they had greatly advanced targeting efficiency so that 56% of injected animals will transmit an edited allele. Similarly, in the Discussion, the authors claim: "As high as 50% of the F_0_ fish could transmit the integrated donor construct to their offspring, which is much higher in germline transmission efficiency than previously reported HR-mediated gene knock-in." Instead, the authors should say they have developed a clever method that makes recovery of conditional alleles very efficient. This method involves first preselecting embryos that have mosaically acquired the conditional allele, and then only screening the preselected transgenic animals for the ability to transmit the alleles through the germline. Under these conditions, up to 56% of the pre-selected founders may transmit edited alleles. The authors should make this clear in the Abstract also.*

Sorry for the confusion in the descriptions about the KI efficiency. We have clarified this issue throughout the section of Results and emphasized the importance of pre-selection of founders to achieve successful germline transmission of KI events in the section of Discussion in our revised manuscript (Results subsections “High efficient generation of a dual-function KI allele at zebrafish *tbx5a* locus” and “Improving the dual-cassette-donor strategy to achieve geno-tagging effect”, and the Discussion section).

2) It is difficult to reconstruct and to figure out the actual numbers that describe the recovery of germline transmissible alleles. For example, let's look at the section that begins subsection "Generation and evaluation of geno-tagging alleles at zebrafish tbx5a locus". The authors write: "After injection and screening, integration of this donor and germline transmission of the tbx5a PoR-NeG donor geno-tagging alleles were detected and confirmed in 2 out of 48 adult F_0_". I can't understand where the number "48" comes from – was this all the F_0_ adults from an injection? Was it 48 F_0_ selected on the basis of fluorescent reporter expression? In Table 2, row 3 it appears that 16/124 injected embryos had reporter expression – so where did the 48 come from?

Sorry for the confusion in the descriptions of our experimental procedures. We have clarified this issue by providing more detailed information about the injection and screening process throughout the section of Results (as summarized in the following) and summarized and discussed the results in the section of Discussion in our revised manuscript. The following is a brief summary:

1) An extract from subsection “Germline transmission and evaluation of the *tbx5a* KI allele”: “To generate *tbx5a* PoR-Ne KI zebrafish, about 100 one-cell stage embryos were injected, and red fluorescent signals could be detected in 18 of them (Supplementary file 2). […] In contrast, no red fluorescent signal was detected in any F1 progeny from outcross of 40 adult F_0_ raised from tdTomato-negative embryos, indicating a positive correlation of reporter gene expression in founder embryos with successful as well as efficient germline transmission of the KI events.”

2) An extract from subsection “Generation and evaluation of dual-function alleles showing CKO with gene labeling effect at zebrafish *kctd10* locus”: “For the generation of kctd10 PoG-Ne KI fish line, we injected about 100 one-cell stage embryos and observed the mosaic expression of tdGFP in 22 F_0_ embryos (Supplementary file 2). We raised all the F_0_ embryos without pre-selection to adulthood, and found 2 of 42 (4.8%) displayed germline transmission screening of 42 F_0_ adults, only two of them displayed germline transmission of the kctd10 PoG-Ne donor KI events (Supplementary file 4).”

3) An extract from subsection “Generation and evaluation of geno-tagging alleles at zebrafish *tbx5a* locus”: “To generate *tbx5a* PoR-NeG geno-tagging zebrafish, about 100 one-cell stage embryos were injected, and 16 of them showed mosaic expression of red fluorescent signals (Supplementary file 2), whose ratio is comparable with the previous *tbx5a* PoR-Ne donor KI experiment. […] These results indicate a strong correlation between germline transmission of KI events and adult somatic tissue integration of the donor, and provide another efficient method for the enrichment of germline transmission events.”

4) An extract from subsection “Geno-tagging effect achieved for zebrafish *sox10* gene”: “To generate the heritable *sox10* PoR-NeG geno-tagging allele, about 150 one-cell stage embryos were injected with the KI system (without Cre mRNA), and the tdTomato fluorescent signals were observed in 103 F_0_ embryos (Supplementary file 2). The positive founder embryos showing fluorescent signals were raised to adulthood and germline transmission screening was conducted by outcross with wild-type zebrafish. Correct red fluorescent patterns in F1 embryos were observed in 10 out of 21 founders (Supplementary file 6).”

3) The authors create conditional alleles that can be switched from WT red to mutant green. They call these PoR-NeG alleles for Positive Red to Negative Green. They also seem to call these geno-tagged alleles. They sometimes breed these to animals with another type of condition allele that switches from a WT-fluorescent reporter to a mutant-no reporter allele. They call these alleles PoNe, positive to negative. It would make the reading much easier if they always labeled an allele that co-expressed a fluorescent reporter. In other words, when the wildtype allele is marked by expression of tdTomato and the mutant allele is not marked by reporter expression, maybe call it PoR-Ne. In addition, they place these PoNe alleles on backgrounds that express a heart reporter (cmcl2:EGFP). As a result, if the WT allele were clearly marked as PoR-Ne then it would be easy to tell which is the heart marker and which is the tagged gene. In general: please use consistent nomenclature.

Sorry for the confusion and thank you for the suggestion. We have clarified this issue as suggested in our revised the manuscript.

4) The figures need some work. The easiest suggestion is that the lettering needs to be larger or at higher resolution – especially for the line drawings – when I expanded it on my screen it became very pixelated. For example, try enlarging Figure 4—figure supplement 1F.

Now we have found a better method to preserve high resolution of our figures, and we believe that the new figures are satisfactory in quality and could display clear details after enlargement.

5) Figure 2A and C: I find these very hard to interpret – I think the authors are visualizing fluorescent markers in three embryo siblings generated from a single cross some of which have been injected with Cre mRNA. The patterns of fluorescence reporter expression differ and I believe the results are to be interpreted as being derived from embryos with different genotypes. Would it be possible for the authors to tell us the presumed genotypes of each row before Cre exposure? I feel this would make the interpretation easier for the reader. I don't understand Figure 2C – it looks like each embryo has a green heart but I think the authors want us to focus on differences. These need to be spelled out perhaps in the figure legend? For Panel E please explain in the figure legend this experiment. Explain the cross and what Normal and Defective embryos are. Explain what "before injection" means.

Sorry for the confusion and thank you for the suggestion. We have clarified these issues by editing the figure and revising the legend as suggested in our new manuscript.

6) Figure 3B: I suspect the markers on the gel are mistakenly labeled. For example, the experimental band of 380 bp is larger than the marker of 500 bp and the experimental band of 820 bp is smaller than the marker of 750 bp. Also, in Panel B, the gel of the 3' junctions: I don't understand how amplification from the donor shown in Panel A with T5F2 and T5R1 would give a band – is there some mistake here?

Sorry for the confusion due to our negligence in gel labeling. We have re-labeled the markers carefully, ensuring they are assigned at the correct position. Regarding the PCR result of the 3' junctions in Panel B, the sequences of both T5F2 and T5R1 primers indeed exist in the donor, flanking the upstream *loxP* site, therefore a 730-bp product could be amplified in the lane with the donor as the template. We have edited the figure and revised the legend to clarify this issue.

7) In Figure 3C, I believe the images of dorsal view and ventral view have been reversed. I think the first column and bottom row picture is a ventral view. Please check – I may be wrong. I think the cross that produced the embryos in Panel C needs to be clearly explained and how to interpret the genotypes should be clearly explained. The images are difficult to interpret – what are we supposed to look at in the boxed insets? Explain in the legend please.

Sorry for the confusion and thank you for the suggestion. We have carefully examined our original microscopic images and confirmed that the dorsal and ventral labeling are correct. We have indicated the expected genotypes in the figure and revised the legend in our new manuscript, as suggested.

8) The images in Figure 3—figure supplement 1 are difficult to interpret. Again I suspect that if the authors told us the predicted genotypes and told us what we are supposed to be seeing in the outlined boxed areas, then it would be clearer.

Sorry for the confusion and thank you for the suggestion. We have clarified these issues by editing the figure and revising the corresponding legend in our new manuscript, as suggested.

[Editors' note: further revisions were requested prior to acceptance, as described below.]

Essential revisions:1) The revised manuscript is much improved, and additional data demonstrating feasibility of fully conditional knockout using the new cmlc2:CreERT2 line. However, one major issue remains. It is common practice to measure the expression of targeted loci in the "off" state in homozygotes. Data can be displayed in several different ways, exemplified by Figure 2 in Ni et al., 2012 reference and Figure 1D in the Grajevskaja et al., 2018. Figure 1A in Sugimoto et al., 2017 serves as a directly relevant example of from a recently published eLife paper describing a conditional knock-in allele. This is very important. As one can appreciate from error bars in Figure 2—figure supplement 1E, Figure 2—figure supplement 2L, Figure 3—figure supplement 1E and Figure 4—figure supplement 1G, it is impossible to distinguish between 90% and 99% mutagenesis efficiency in heterozygotes (55% vs. 51% of read-through transcript remaining in heterozygotes). In contrast, the difference between 10% and 1% levels of remaining wild-type transcript should be quite straightforward to reliably quantify in homozygotes. Results of such quantitative analysis may have an impact on the choice between your described polyA cassettes (tandem SV40/GBH vs. 8xSV40). Thus, qRT-PCR analyses must be performed on embryos homozygous for Cre-excised alleles.

Thank you for the suggestions. We have measured the expression of all the four targeted loci in the “off” state from their corresponding homozygous mutant embryos by qRT-PCR. The results showed that the wild-type transcripts displayed at least a 12.5-fold reduction in the homozygous mutant embryos for each targeted allele. More precisely, their transcription level decreased to about 6.4%, 0.3%, 7.9% and 4.1% for *tbx5a^Ne/Ne^* (Figure 2—figure supplement 1F; subsection “Conditional knockout could be achieved for the *tbx5a* KI allele”), *kctd10^Ne-1/Ne-1^* (Figure 2—figure supplement 2M; subsection “Generation and evaluation of dual-function alleles showing CKO with a gene labeling effect at the zebrafish *kctd10* locus”), *tbx5a^NeG/NeG^* (Figure 3—figure supplement 1F; subsection “Geno-tagging effect achieved for the zebrafish *sox10* gene”), and *sox10^NeG/NeG^* (Figure 4—figure supplement 1H; Discussion section), respectively.

2) A second issue somewhat detracting from the overall quality of the work is occasionally difficult-to-read English. I would simply recommend that the final submission should be edited with the help of a professional editor.

Thank you for the suggestions. Our current manuscript has been polished by the language editing service team of Nature Publishing Group, and we hope the new revision is now satisfactory.

3) Sequences of primers used for qRT-PCR should be included, along with other primers, in Supplementary file 7.

Sorry for the confusion. The sequences of primers used for PCR and qRT-PCR analyses were previously listed in the Supplementary file 7 and Supplementary file 8, respectively. Now we have combined all the primer sequence information into a single table as the new Supplementary file 7.

4). Why is kctd10 deleted from the Abstract?

Sorry for the confusion. We deleted *kctd10* from the abstract due to word limitation. Now we have revised the Abstract and added *kctd10* back to the new Abstract.

5) Subsection “High efficient generation of a dual-function KI allele at zebrafish tbx5a locus”: Need to show hEMX1 target site on the vector in Figure 1A, and corresponding sites in vector diagrams in subsequent figures.

Sorry for the confusion. We actually provided the *hEMX1* target site sequence on top of the donor vector, and depicted the site as a purple box in the diagram of the donor vector in our previous Figure 1A. Now we have improved the figure by adjusting the color of the box to better match the color of the above *hEMX1* target site sequence, and labeling the box by adding the word “*hEMX1* site” below it, as well as labeling the donor at its upper-left corner with “Donor vector”.

Similarly, we have also revised all the other figures containing vector diagrams, including Figure 2—figure supplement 2D and Figure 3A for the *hEMX1* site, as well as Figure 4—figure supplement 1D for the *lamGolden* site.